# Paired DNA and RNA sequencing uncovers common and rare variation regulating human retinal gene expression

Jacob Sampson[1], Ayellet V. Segrè [2,3,4], Kinga M. Bujakowska [2,3], Simon J. Clark[1,5,6], Paul N. Bishop[1], Steve Haynes[1], Diana Baralle[7], Jospin Al-Deek[1], Stacey Holden [1], Beverley Anderson[1], Andrew Hayes [1], Rahmat A. Kemal [1,8], Huw B. Thomas [1], Raymond T. O'Keefe [1], Siddharth Banka [1,9], Graeme C. Black [1,9], Panagiotis I. Sergouniotis [1,9,10,11] & Jamie M. Ellingford [1] ✉

Genetic disorders impacting vision affect millions of individuals worldwide, including age-related macular degeneration (common) and inherited retinal disorders (rare). There is an incomplete understanding of the impact of genetic variation on gene expression in the human retina and its role in genetic disorders. Through the generation of whole genome sequencing and bulk RNA-sequencing of neurosensory retina and retinal pigment epithelium from 201 post-mortem eyes, we uncover common and rare genomic variants shaping retinal expression profiles. This includes 1,483,595 significant cis-expression quantitative trait loci impacting 9,959 and 3,699 genes in neuro-sensory retina and retinal pigment epithelium, respectively, with associated genomic variants enriched to cis-candidate regulatory elements and notable shared eGenes between both tissues. We also detect 1051 expression outliers and prioritise 299 rare non-coding single-nucleotide, structural variants or copy number variants as plausible drivers for 28% of outlier events. This study increases understanding of gene expression regulation in the human retina.

Genomic variation has been well established to play a role in the onset and susceptibility of visual impairment by disrupting normal functioning of the retina, a highly specialised light-sensitive tissue at the back of eye[1]. The retina depends on the interaction between neuronal and non-neuronal cell types, including those in the neurosensory retina (NSR), e.g. photoreceptors and ganglion cells[2] and the retinal pigment epithelium (RPE), a monolayer which lines the photoreceptor outer segments[3]. Inherited retinal disorders (IRDs) are a diverse set of largely monogenic conditions driven primarily by highly impactful genetic variants that are rare in the population and disrupt the function of the NSR and/or RPE[4]. Monogenic IRDs may present in isolation, for example, Stargardt disease, retinitis pigmentosa and cone-rod dystrophy or as part of a multi-system disorder, for example, Usher syndrome, Joubert syndrome and Senior-Loken syndrome. Age-related

[1]Division of Evolution, Infection and Genomics, School of Biological Sciences, Faculty of Biology, Medicine and Health, University of Manchester, Manchester, UK. [2]Ocular Genomics Institute, Department of Ophthalmology, Massachusetts Eye and Ear, Boston, MA, USA. [3]Department of Ophthalmology, Harvard Medical School, Boston, MA, USA. [4]Broad Institute of Harvard and MIT, Cambridge, MA, USA. [5]University Eye Clinic, Eberhard Karls University of Tübingen, Tübingen, Germany. [6]Institute for Ophthalmic Research, Eberhard Karls University of Tübingen, Tübingen, Germany. [7]School of Human Development and Health, Faculty of Medicine, University of Southampton, Southampton, UK. [8]Department of Medical Biology, Faculty of Medicine, Universitas Riau, Pekanbaru, Indonesia. [9]Manchester Centre for Genomic Medicine, Manchester University NHS Foundation Trust, Health Innovation Manchester, Manchester, UK. [10]European Molecular Biology Laboratory, European Bioinformatics Institute (EMBL- EBI), Wellcome Genome Campus, Cambridge, UK. [11]Manchester Royal Eye Hospital, Manchester University NHS Foundation Trust, Manchester, UK. ✉e-mail: jamie.ellingford@manchester.ac.uk

macular degeneration (AMD) is a common disorder that impacts the retina and is a leading cause of visual impairment in adults[5], predicted to impact 288 million individuals by 2040[6]. Whilst non-genetic risk factors exist for AMD, including age, diet and lifestyle, its heritability is estimated to be as high as 71%[7]. Genome-wide association studies (GWAS) initially identified more than 50 genomic loci impacting 34 genes that convey high risk to AMD in a European ancestry cohort[8]. Recent expansion of AMD GWAS to Hispanic and African ancestries has uncovered 30 additional genomic loci and distinct AMD genomic architecture in these populations[9].

The Genotype-Tissue Expression (GTEx) project has transformed our ability to pinpoint genetic variants that impact gene expression[10], including tissue-specific and tissue-shared expression quantitative loci (eQTLs) and rare genetic variants associated with expression outlier (eOutlier) events. Findings from these investigations, along with other studies, have been leveraged across various biological and medical fields to gain a deeper understanding of disease mechanisms[11,12], to provide more informed diagnosis and prognosis[13] and to pursue pathways for novel treatments[14]. Notably, ophthalmic tissue was not included in the GTEx resource, but recently, the EyeGEx study identified over 2 million eQTLs in the retina, from a cohort comprised of healthy eyes and eyes displaying signs of AMD[15] and Strunz et al.[16] identified 580,171 eQTLs in the neural retina from a cohort of healthy eyes. These resources enable the investigation of the role of common single-nucleotide variants (SNVs) influencing retinal gene expression. However, to our knowledge, there are no suitable datasets to also interrogate the impact of rare variants, structural variants (SVs) and copy number variants (CNVs) on retinal gene expression. Expanding our understanding of gene expression regulation in the retina will provide insights into the molecular mechanisms underlying both common and rare eye diseases and help identify new potential strategies for treatment and prevention.

Here, we describe the creation of a paired genomic and transcriptomic resource for the human retina from 201 donors and develop a new understanding of both common and rare variants that drive expression in this highly specialized tissue.

## Results

### The METR genome-transcriptome resource integrates genomic and retinal transcriptomic data from 201 post-mortem eye samples

The Manchester Eye Tissue Repository (METR) genome-transcriptome cohort comprises 201 unrelated individuals who donated eye tissue post-mortem. The median age of the cohort was 71 years (IQR 64–77) at the time of post-mortem, with a slight male predominance (63.7%). The median ischemic time was 40 h (IQR = 32–44) (Supplementary fig. 1). While 47 individuals (23% of the cohort) were found to carry genetic variants that confer high-risk for age-related macular degeneration (AMD), none of the 201 individuals included in the cohort had phenotypic presentation, assessed post-mortem, consistent with late-stage AMD or monogenic ophthalmic disorders.

Short-read whole genome sequencing was performed on an Illumina NovaSeq6000, with alignment and variant detection performed using DRAGEN software (v4.0.3). The median genome-wide average coverage per sample was 35.9x (IQR = 30.3–40.5) (Supplementary fig. 2), with an average of 88.0% and 92.8% of the genome covered by at least 15 or at least 10 sequencing reads, respectively. Joint SNV calling with DRAGEN PopGen 4.2.4 obtained aggregate calls at 15,617,784 high-confidence variant sites after quality control (Supplementary Data 1). On average, 173 CNVs and 8,814 SVs were identified per sample (Supplementary Data 2). Genomic variation profiles among the 201 donors confirmed that the cohort was exclusively of European genetically inferred ancestry (Supplementary fig. 3).

Transcriptomic data were generated by short-read bulk RNA sequencing of polyadenylated enriched RNA, using an Illumina NovaSeq6000. The median RNA integrity number (RIN) for samples selected for transcriptomic analysis was 7.9 (IQR = 7.5–8.1) for NSR samples ($n = 183$) and 6.9 (IQR = 6.5–7.5) for RPE samples ($n = 176$) (Supplementary figs. 1 and 4A and Supplementary Data 3). We obtained an average of 139 million uniquely mapped reads for NSR (IQR = 138–161 million) and 62.3 million for RPE (IQR = 59–94 million), representing 89.4% (IQR = 88.2–90.7%) and 82.9% (IQR = 78–86%) of all generated reads, respectively (Supplementary fig. 4). The median 3'/5' bias, defined as the ratio of sequencing depth between the 150 bp region at the 3' end and the 5' end of the gene for genes with a length greater than 600 bp and at least 5 unambiguous reads, was 0.5 for NSR samples (IQR = 0.48–0.51) and 0.51 for RPE samples (IQR = 0.49–0.55) (Supplementary fig. 1B). Some level of expression (mean TPM > 0.1) was indicated for 28,512 genes across both tissues, with 18,891 and 13,214 genes expressed at moderate (TPM > 1) and high (TPM > 5) levels, respectively (Fig. 1a). 59% of expressed genes (mean TPM > 0.1) ($n = 16,785$) and 90% of highly expressed genes (mean TPM > 5) ($n = 11,984$) were protein coding, representing 84% and 60% of all GENCODE protein coding genes, respectively. Significantly higher expression variability was observed in the RPE compared to the NSR for genes expressed at low, moderate and high levels in samples with both NSR and RPE data, measured by the coefficient of variation ($p$-value $< 2.2 \times 10^{-16}$; Fig. 1b).

To ensure the validity of the transcriptomic datasets generated in this study, we assessed the biological relevance of expressed genes in NSR and RPE. Gene expression profiles were enriched for gene ontology (GO) terms indicative of the tissues of origin (Fig. 1c). Overall, 14,957 differentially expressed genes (mean expression > 1 TPM and adj. $p$-value < 0.05) were identified between the RPE and the NSR. Unsurprisingly, cell type deconvolution analyses, with reference to single-cell retinal datasets[17], demonstrated a significantly higher representation of RPE cells in data generated from RPE samples compared to data generated from NSR samples (Supplementary fig. 5A). Moreover, genes with increased expression in the RPE ($n = 7353$) were enriched for 987 GO terms, which were grouped into 55 non-redundant clusters, including epithelial cell proliferation (GO:0050678), regulation of cell adhesion (GO:0030155) and positive regulation of immune response (GO:0050778) (Supplementary Data 4). Deconvolution of the NSR datasets supported the presence of at least 7 neuronal cell types at high levels (estimated proportion > 1%), with an average relative composition, per sample, of 29% rod photoreceptors (95%CI:26.4-30.7%), 28% retinal astrocytes (95% CI = 26.1–29.6%), 16% amacrine cells (95% CI = 15.7–17.1%), 10% horizontal cells (95% CI = 9.5–10.2%), 7% retinal ganglion cells (95% CI = 5.8–9%), 4% bipolar cells (95% CI = 3.8–4.4%), 2% Müller glia (95% CI = 1.7–3.1%) and ~4% other cell types (Supplementary fig. 5C). Genes with increased expression in the NSR, compared to the RPE ($n = 7604$), were enriched for 238 GO terms, grouped into 27 non-redundant clusters including synapse organisation (GO:0050808), neurotransmitter transport (GO:0006836) and cell morphogenesis involved in neuron differentiation (GO:0048667) (Supplementary Data 4).

### METR eQTLs provide novel insights into non-coding variants that impact known eye disease-related genes

We performed cis-eQTL mapping to identify common genetic variants that are associated with gene expression in the NSR and the RPE. We found 1,424,946 significant (FDR < 0.05) cis-eQTL associations between 806,789 variants (eVariants) and 9959 genes (eGenes) in NSR (Supplementary fig. 6). Additionally, 465,045 eQTLs were identified between 303,773 eVariants and 3699 eGenes in the RPE (Supplementary fig. 6). The lower range of alternate internal allele frequencies for eVariants identified as part of eQTLs was 2.5% and included novel variants in comparison to gnomAD v4.1 (Supplementary fig. 7). 406,396 eQTLs were common to both the retina and the RPE, while 1,018,550 associations were NSR-specific and 58,649 were RPE-specific

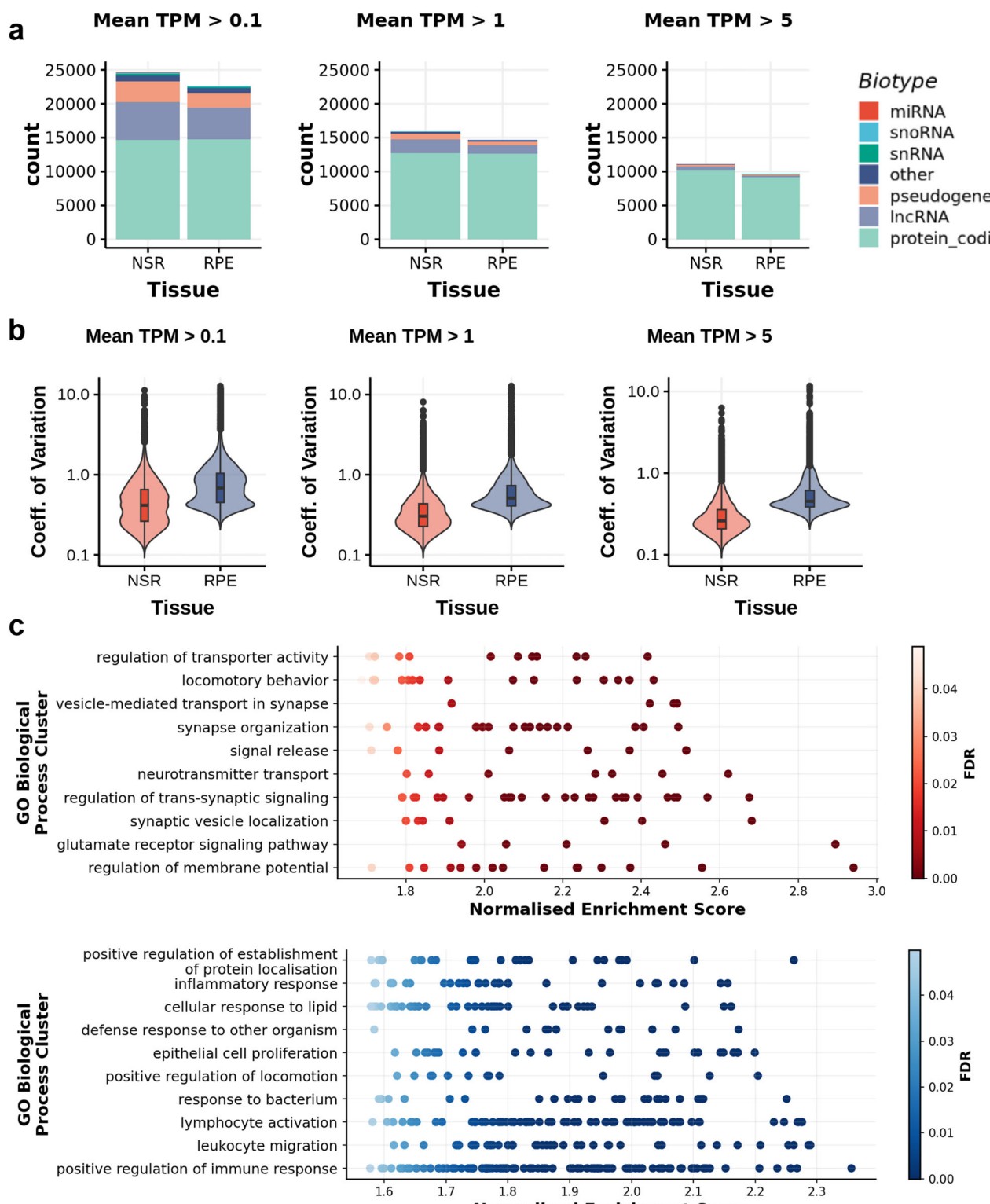

**Fig. 1 | Tissue-specific transcriptomic data were generated for the neurosensory retina (NSR) and the retinal pigment epithelium (RPE). a** Number of genes expressed in the NSR and the RPE at different expression thresholds, classified into different biotypes. **b** The coefficient of variation (mean/SD) for all genes expressed at different thresholds in both tissues indicates higher expression variability across samples in the RPE (*n* = 176) compared to the NSR (*n* = 183). Box plots show the median (centre line), interquartile range (box) and minimum and maximum values (whiskers). **c** Top 10 Gene Ontology Biological Process clusters enriched in differentially expressed genes (adj pvalue < 0.01) in the NSR and RPE, respectively. Gene Ontology terms (FDR < 0.01) were sorted by enrichment ratio and clustered based on semantic similarity.

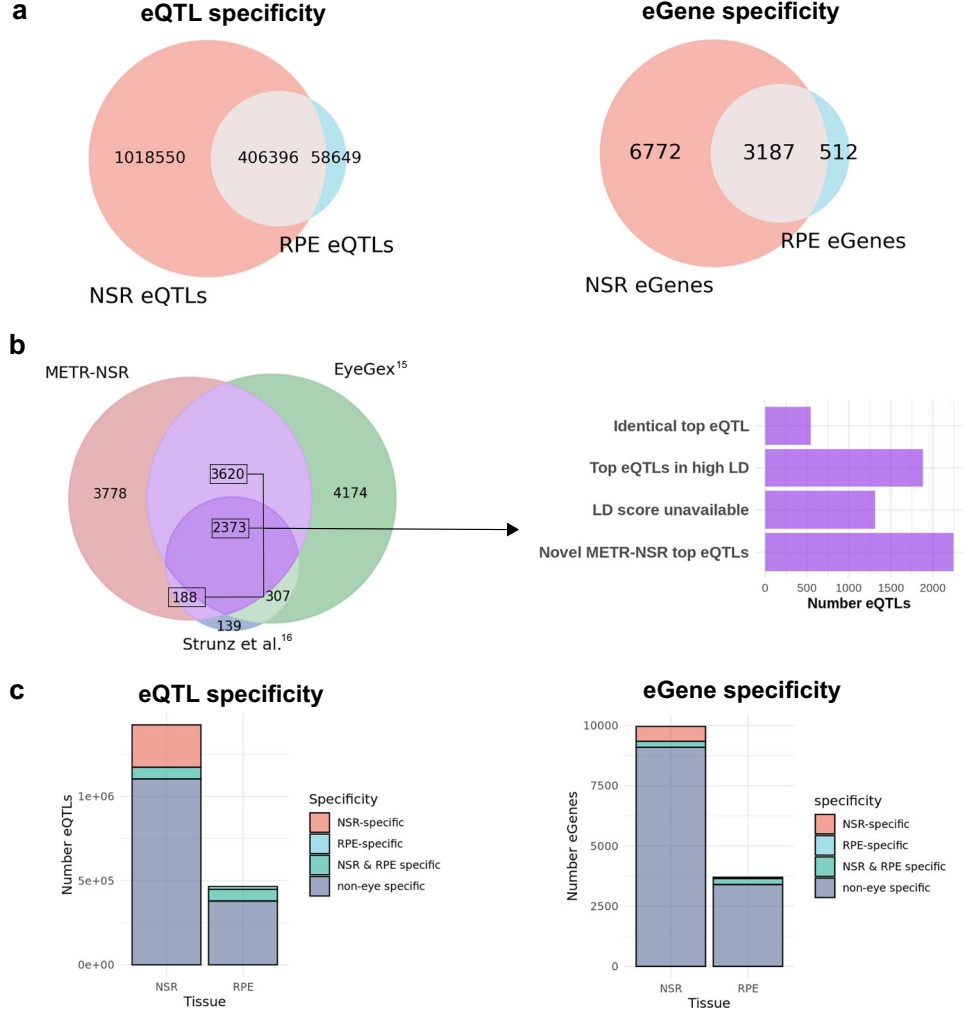

**Fig. 2 | Intersection between METR-eQTLs and eQTLs from different studies.**
**a** Overlap between METR-eQTLs (left) and METR-eGenes (right) identified in the neurosensory retina (NSR) and the retinal pigment epithelium (RPE). **b** Intersection between eGenes identified in the neurosensory retina (METR-NSR), EyeGex[15] and Strunz et al.[16] retina eQTL study. For eGenes which were present in METR-NSR and at least one other study, we compared the top eQTL for each eGene to identify 1)

eQTLs which were replicated in an additional study/studies; 2) eGenes where the top eQTL from each study was in high LD with each other (r2 > 0.8) and 3) eGenes where the top NSR hit was novel. The LD score was unavailable for a subset of eQTLs. **c** Comparison between METR-eQTLs and GTEX eQTLs indicates the number of NSR-specific, RPE-specific, NSR & RPE-specific and non-eye-specific eQTLs (left) and eGenes (right) from both tissues in our study.

(Fig. 2a). Henceforth, we will refer to eQTLs identified in the NSR and/or the RPE as METR-eQTLs ($n = 1,483,595$), which included 10,471 unique eGenes (6772 NSR-specific, 512 RPE-specific and 3187 eGenes in both NSR and RPE).

We compared the top eQTLs identified for NSR (METR-NSR eQTLs) for each eGene identified in this study with two published retina-specific eQTL datasets, the EyeGEx project[15] and Strunz et al.[16], to identify: (1) eQTLs identically replicated in the NSR tissues; (2) METR-NSR eQTLs that impacted eGenes previously described but had alternative eVariants in high linkage disequilibrium (LD) with findings from EyeGEx or Strunz et al. and (3) previously unreported eQTLs for NSR, including newly identified eGenes (Fig. 2b and Supplementary fig. 8). Of note, our cohort excludes individuals with late-stage AMD, whereas EyeGEx includes late-stage AMD eyes. We report 6181 NSR eGenes which were previously described by at least one other study (62% of all NSR eGenes), of which 547 eGenes (9%) share identical top eVariants with at least one other study and 1882 (30%) have top eVariants in high LD with previously identified top eVariants ($r^2 > 0.8$) (Fig. 2b). We identified 343,527 novel eQTLs (24%) in eGenes that were previously described by at least one previous study and 386,741 novel eQTLs (27%) in 3,778 newly identified eGenes. Importantly, we

replicate 13 eQTLs that have previously been reported to impact genes that are implicated in increased risk of AMD (Table 1).

**Over 800 eGenes are newly identified in the NSR and RPE.** To evaluate the tissue-specificity of our dataset, we compared the METR-eQTLs with non-eye-specific eQTLs from the GTEx project (Fig. 2c). We identified 337,424 METR-eQTLs (22.7%) and 916 eGenes (8.7%) that had not been previously identified by GTEx (Fig. 2c); 251,685 (74.6%) of these eQTLs have not been previously described as eQTLs in NSR or RPE previously. Of the novel eGenes, 479 (57.9%) encoded lncRNAs, and 5 had previously been associated with a known rare monogenic eye disease (*HPS4, ACO2, CRX, CRYAA, PEX26*). We evaluated the degree of similarity between METR-eQTLs and eQTLs from each GTEx tissue using the Intersection over Union (IoU) statistic, which accounts for the wide variation in the number of eQTLs from different tissues (Supplementary Figs. 9 and 10). The brain cortex had the highest level of eQTL similarity to our dataset (IoU = 0.28) and 5 of the top 10 most similar tissues were from the brain.

**Genetic variants driving expression profile differences are enriched in candidate cis-regulatory elements (cCREs), with the**

**Table 1 | Replication of eQTLs that impact AMD risk genes and were identified as lead candidates by Ratnapriya et al.[15] and Orozco et al.[26] in the METR-eQTL dataset**

| AMD risk locus | Gene ID | Reported candidate eVariant from previous study | Source | Category | Overlapping regulatory elements (tissue; type) |
|---|---|---|---|---|---|
| *ACAD10* | *SH2B3* | chr12_111694806_G_A | EyeGex[15] | Novel METR eQTL | |
| *ARMS2/HTRA1* | *HTRA1* | chr10_122534138_C_T | Orozco et al.[26] | Novel METR eQTL | |
| *B3GALTL* | *B3GLCT* | chr13_31247103_C_T | EyeGex[15] | Not replicated | Non-retina; enhancer |
| *B3GALTL* | *B3GLCT* | chr13_31133338_T_A | Orozco et al.[26] | Novel METR eQTL | |
| *CFI* | *PLA2G12A* | chr4_109737911_T_C | EyeGex[15] | Novel METR eQTL | |
| *CFI* | *CFI* | chr4_109594995_C_G | Orozco et al.[26] | No METR eQTL | |
| *COL4A3* | *COL4A3* | chr2_227254233_C_G | Orozco et al.[26] | No METR eQTL | |
| *CTRB2/CTRB1* | *BCAR1* | chr16_75208831_T_C | Orozco et al.[26] | METR eQTL in high LD | |
| *MMP9* | *SLC12A5-AS1* | chr20_46006318_C_T | Orozco et al.[26] | Replicated | Non-retina; enhancer; |
| *NPLOC4-TSPAN10* | *TSPAN10* | chr17_81654604_A_G | Orozco et al.[26] | Replicated | |
| *PILRB/PILRA* | *PILRB* | chr7_100393925_C_T | EyeGex[15] | Replicated | Retina and non-retina, enhancer, scATACseq (cones, bipolar cells |
| *PILRB/PILRA* | *PILRA* | chr7_100393925_C_T | EyeGex[15] | Replicated | Retina and non-retina, enhancer, scATACseq (cones, bipolar cells) |
| *PILRB/PILRA* | *ZCWPW1* | chr7_100393925_C_T | EyeGex[15] | Replicated | Retina and non-retina, enhancer, scATACseq (cones, bipolar cells) |
| *PILRB/PILRA* | *TSC22D4* | chr7_100393925_C_T | EyeGex[15] | Replicated | Retina and non-retina, enhancer, scATACseq (cones, bipolar cells) |
| *PILRB/PILRA* | *PILRB* | chr7_100375779_G_A | Orozco et al.[26] | Replicated | |
| *PILRB/PILRA* | *PILRA* | chr7_100345960_A_T | Orozco et al.[26] | Replicated | Non-retina; enhancer |
| *RDH5/CD63* | *BLOC1S1* | chr12_55721994_C_A | EyeGex[15] | Replicated | Non-retina; enhancer |
| *RDH5-CD63* | *BLOC1S1* | chr12_55819513_G_T | Orozco et al.[26] | Replicated | |
| *RDH5-CD63* | *RDH5* | chr12_55721801_C_T | Orozco et al.[26] | No METR eQTL | Non-retina; enhancer |
| *SLC16A8* | *BAIAP2L2* | chr22_38071777_G_C | Orozco et al.[26] | No METR eQTL | Non-retina; enhancer |
| *TMEM97/VTN* | *POLDIP2* | chr17_28322698_A_C | EyeGex[15] | Replicated | Non-retina; enhancer |
| *TMEM97/VTN* | *SLC13A2* | chr17_28322698_A_C | EyeGex[15] | Novel METR eQTL | Non-retina; enhancer |
| *TMEM97/VTN* | *TMEM199* | chr17_28322698_A_C | EyeGex[15] | Replicated | Non-retina; enhancer |
| *TMEM97/VTN* | *TMEM199* | chr17_28376663_T_A | Orozco et al.[26] | Replicated | |
| *TNFRSF10A* | *TNFRSF10A* | chr8_23231471_C_G | Orozco et al.[26] | No METR eQTL | Non-retina; enhancer |
| *TRPM1* | *TRPM1* | chr15_31080689_T_C | Orozco et al.[26] | METR-eQTL in high LD | Non-retina; enhancer |

26 eQTLs across 14 AMD risk loci have previously been identified. Of these, 15 were either replicated (n = 13) or in high LD (n = 2) (r$^2$ > 0.8) with METR-eQTLs. Novel METR-eQTLs are identified for 5 genes previously implicated in AMD risk. 10 of the 21 (48%) unique eVariants overlap with characterised candidate cis-regulatory elements, although only 1 of these has previously been characterised as active in retinal cells. Definitions utilised in table: 'Replicated', lead candidate variant from other studies is also identified in this study; 'METR eQTL in high LD', lead candidate variant from other studies is in high LD with an eQTL identified in this study; 'Novel METR eQTL', the lead candidate variant from other studies is not replicated in this study, additional and previously unreported eVariants are identified in this study but they are not in high LD with the previously reported lead candidate variant; 'No METR eQTL', the gene is not identified as an eGene in this study within the statistical parameters applied (FDR < 0.05); 'Not replicated', we identify eQTLs for the gene in this study but the lead candidate variant from other studies is not replicated nor is it in high LD with an eQTL identified in this study.

**highest enrichment in retina-specific cCREs.** To understand whether eQTLs were enriched for putative regulatory regions, we compared locations of METR-eVariants to cell-type agnostic cis-candidate regulatory elements (cCRE) available through ENCODE (V3). METR-eVariants were enriched in cell-type agnostic promoters ($p = 8.05 \times 10^{-19}$) and proximal enhancers ($p = 8.48 \times 10^{-26}$), compared to control variants matched for allele frequency and gene density. There was no enrichment of eVariants in distal enhancers, CTCF binding sites or DNase-H3K4me3 sites (Fig. 3a).

When stratified by cell-specific regulatory regions, bootstrapping analysis indicated a significant enrichment of METR-eVariants in NSR-specific ($p$-value = $4.52 \times 10^{-28}$) and RPE-specific cCREs ($p$-value = $8.74 \times 10^{-10}$)[18] compared to control variants matched for allele frequency and gene density (number of gene TSSs within 1 Mb of variant) (Fig. 3b). Furthermore, we observe a significant enrichment of METR-eVariants in cell-type specific accessible chromatin regions across 8 different retina cell types[19], with the greatest enrichment in rod cells ($p$-value = $6.69 \times 10^{-58}$) and cone cells ($p$-value = $6.58 \times 10^{-57}$) (Fig. 3c). Non-eye cCREs from adult tissues in EpiMap were also

enriched for METR-eVariants, although the enrichment was lower than in the NSR and RPE. Despite the relative enrichment in annotated regulatory loci, most METR-eVariants (88.2%) do not overlap with any previously characterised cCREs (Fig. 3d).

Further, we assessed whether eVariants previously implicated in AMD risk intersected with cell-type agnostic or cell-specific regulatory regions, observing overlap for 48% (10/21) of unique eVariants with characterised cis-regulatory elements, although only 1 eVariant overlapped with regions previously shown to be active in the retina (Table 1).

**Properties of METR-eQTLs differ between known monogenic disease genes and non-disease-related genes.** To understand if there were trends that were specific to eQTLs associated with known monogenic eye disease genes, we compared findings from this study against the EyeG2P resource[20]. We identified 230 METR-eGenes that were described as causes of rare monogenic disorders in EyeG2P (eye-disease genes) and compared trends identified in these genes against all other METR-eGenes (n = 10,241) (eye non-disease genes) (Fig. 4). We

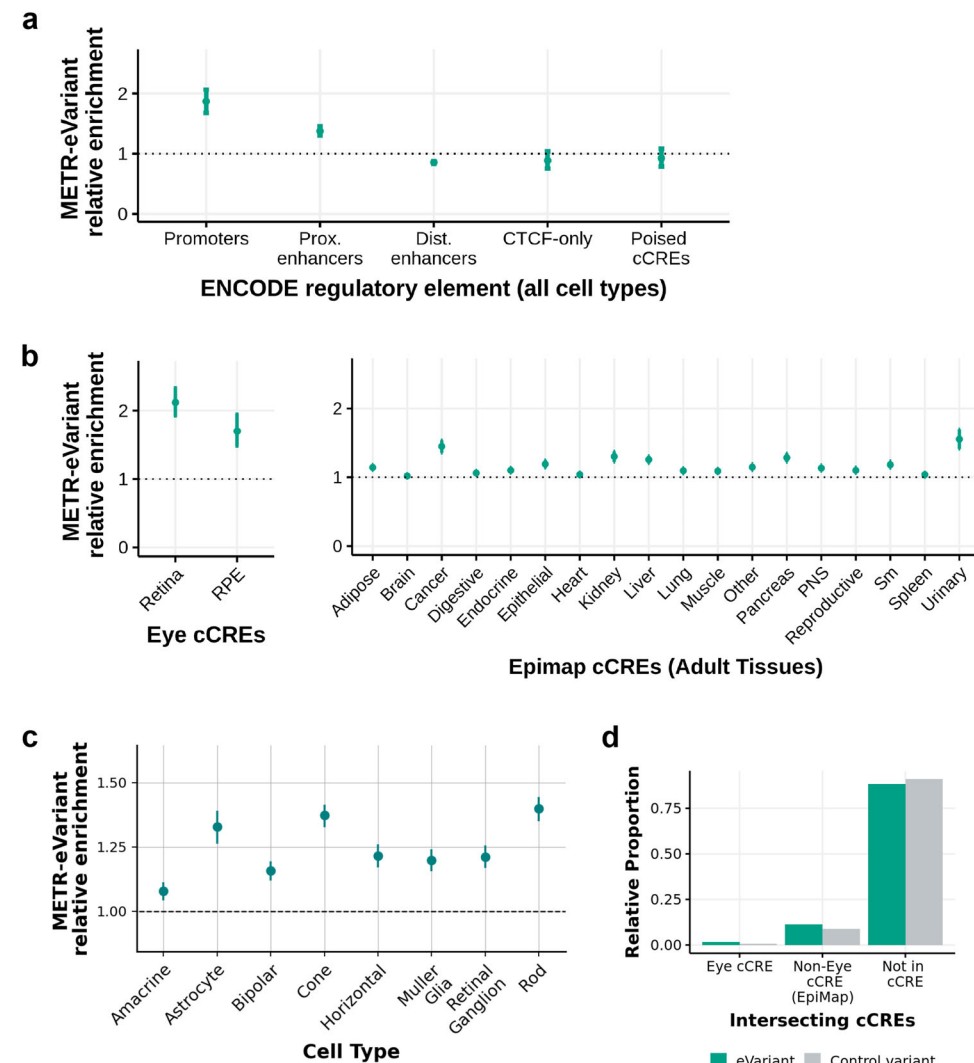

**Fig. 3 | Enrichment of eye eVariants in previously annotated candidate cis-regulatory elements (cCREs). a** Bootstrapped relative enrichment of eye eVariants ($n = 1,483,595$) which intersect with cCREs from the ENCODE cCRE registry (V3)[83] compared to control variants ($n = 6,041,074$). Relative enrichment is defined as the ratio of eVariants to control variants that intersect with each element. In each bootstrapping iteration, random subsets of eVariants and control variants (subsets of non-eVariants matched for gnomad AF and gene density) were intersected with each cCRE group. Centre points indicate the mean and error bars indicate the 2.5–97.5% confidence intervals. **b** Relative enrichment of eye eVariants ($n = 1,483,595$) which intersect with cCREs from retina and RPE from Cherry et al.[18] and cCREs from adult tissues in EpiMap[23] compared to control variants matched for gnomad AF and gene density ($n = 6,041,074$). Points indicate the mean and error bars indicate the 2.5-97.5% confidence intervals **c** Relative enrichment of eVariants ($n = 1,483,595$) compared to control variants ($n = 6,041,074$), which intersect accessible chromatin regions from single cell ATACseq peaks from Wang et al.[19] in different retina cell types. Points indicate the mean and error bars indicate the 2.5–97.5% confidence intervals **d** Relative proportion of eVariants ($n = 1,483,595$) and non-eVariants ($n = 6,041,074$) in the NSR and/or RPE, which intersect with annotated retina cCREs from Cherry et al.[18], non-retina cCREs (EpiMap[23]) and neither. Control variants refer to those that were included in eQTL mapping (MAF > 2.5% and AC > 10) but were not associated with any eQTLs in NSR or RPE.

observed significantly lower expression variability across samples for eye disease eGenes compared to eye non-disease eGenes ($p < 2.2 \times 10^{-16}$) (Fig. 4a). Eye disease eGenes were associated with significantly fewer eQTLs per gene ($p = 7.8 \times 10^{-3}$) (Fig. 4b). Additionally, eQTLs associated with eye disease eGenes have a significantly lower impact on gene expression ($p < 2.2 \times 10^{-16}$) (Fig. 4c) and significantly higher allele frequency (AF, gnomAD v4) compared to eye non-disease eVariants ($p < 2.2 \times 10^{-16}$) (Fig. 4d). Genes that have been associated with rare monogenic eye disease have higher expression (mean TPM = 37.9) than non-disease genes (mean TPM = 20.3) and to control for this potential confounding factor, we adopted a bootstrapping approach ($n = 1000$ iterations) to randomly resample 100 eQTLs associated with eye-disease genes and 100 eQTLs associated with non-eye disease genes matched for gene expression level ($\pm 5\%$ TPM) (Supplementary fig. 11). The direction of trends remained similar after

bootstrapping, with lower effect sizes and higher allele frequencies observed for eQTLs associated with eye disease genes than non-disease genes (Supplementary fig. 11). For both eye disease eGenes and eye non-disease eGenes, there is a negative correlation ($p < 2.2 \times 10^{-16}$) between eVariant allele frequency and the impact of each eQTL on gene expression (Fig. 4E). These findings are consistent with the hypothesis that eVariants which are more common in the population have lower effect sizes on gene expression compared to rarer eVariants (min eVariant allele frequency = 2.5%) and are suggestive of a selective bias against rarer eVariants impacting known eye disease genes.

## Rare variants are plausible drivers of transcriptomic outliers in NSR and RPE

We utilised the DROP workflow[21] to identify statistical outlier events within the METR transcriptome datasets, including expression,

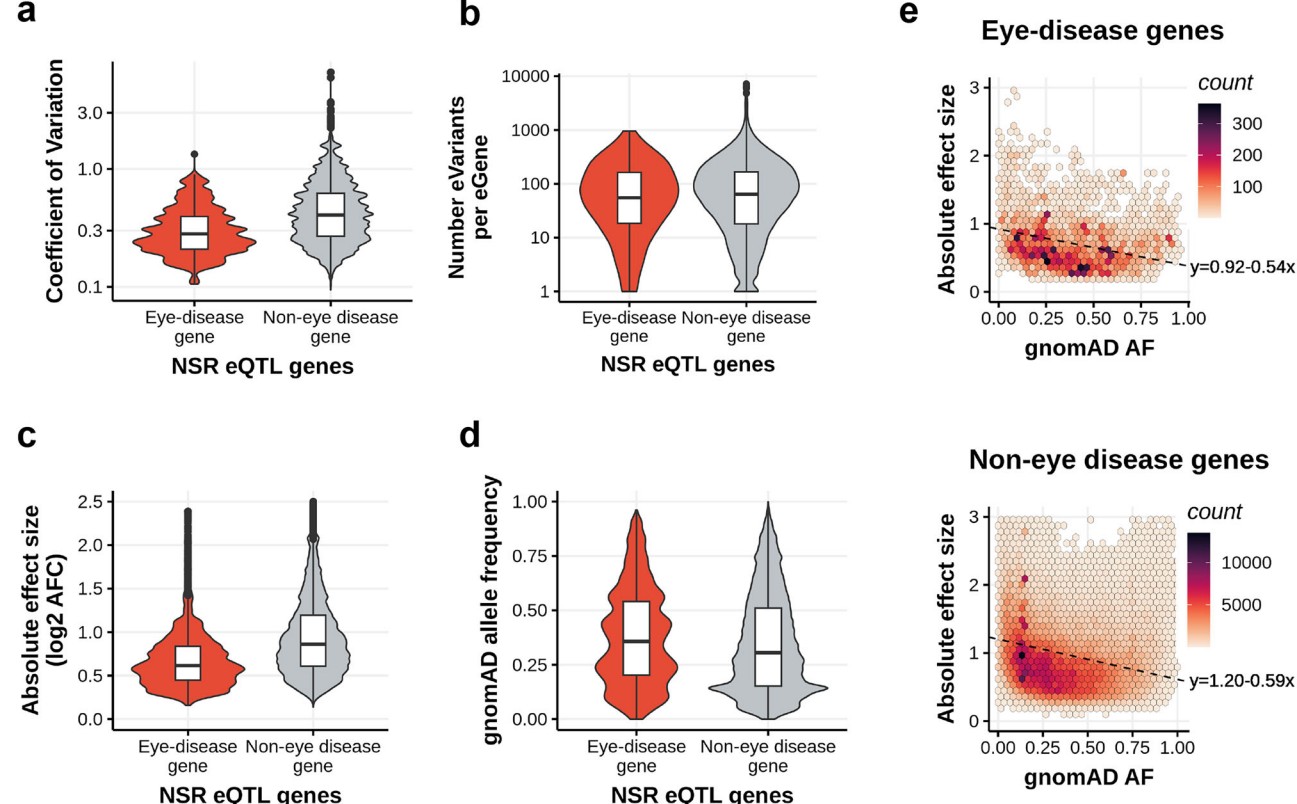

**Fig. 4 | Differences in NSR eQTLs associated with known rare monogenic eye disease genes and genes which are not attributed to eye diseases (non-eye disease genes). a** NSR eGenes associated with eye diseases on EyeG2P (n = 215) have lower coefficients of variation than non-eye disease eGenes (n = 9744), indicating lower expression variability across samples (p < 2.2 × 10⁻¹⁶). **b** On average, known eye disease eGenes (n = 215) have fewer eVariants per gene than non-eye disease eGenes (n = 9744) (p = 7.8 × 10⁻³). **c** The impact on gene expression (measured in absolute log2 allelic fold change) of each eQTL associated with a known eye disease gene (n = 25,205) is lower than the impact of eQTLs associated with non-eye disease genes (n = 1,425,748) (p < 2.2 × 10⁻¹⁶). **d** The allele frequency on gnomAD for the eVariants associated with eye disease genes (n = 24,674) is higher than non-eye disease genes (n = 794,167) (p < 2.2 × 10⁻¹⁶). **e** There is a negative linear relationship between the allele frequency of a variant and the impact on gene expression of the associated eGene (measured in log2(allelic fold change)) for eQTLs associated with eye disease genes (top) and non-eye disease genes (bottom). All reported p-values (**: p < 0.01; ***: p < 0.001) were calculated with a two-sided independent t-test. All box plots (**a–d**) show the median (centre line), interquartile range (box) and minimum and maximum values (whiskers).

### Table 2 | Transcriptome outliers detected in the METR-cohort

| Type of outlier | Expression | | | Splicing | | | Allelic balance | | |
|---|---|---|---|---|---|---|---|---|---|
| Tissue | NSR | RPE | Shared | NSR | RPE | Shared | NSR | RPE | Shared |
| Unique events (count) | 728 | 443 | 120 | 8610 | 8923 | 469 | 37,230 | 106,116 | 9888 |
| Unique genes (count) | 702 | 439 | 137 | 6095 | 6270 | 2972 | 7893 | 17,167 | 7410 |
| Number of genes identified as outliers, per sample (median) | 3 | 1 | 0 | 12 | 6 | 1 | 173 | 138 | 53.5 |
| IQR (Q1, Q2) | (1, 4) | (0,2) | (0, 1) | (8,18) | (3, 24) | (0,2) | (146, 203) | (71, 1003) | (34, 87) |
| Range (min, max) | (0, 64) | (0, 46) | (0, 24) | (2, 1948) | (0, 1258) | (0, 209) | (102, 2823) | (14, 4308) | (8, 214) |

The DROP pipeline was utilised to identify three types of transcriptome outliers (expression, splicing and monoallelic expression) in the NSR and RPE.

splicing and allelic imbalance outliers (Table 2). We identified 1,051 unique instances of a gene being aberrantly expressed in an METR sample (METR expression outlier events, METR-eOutlier events) (adjusted p < 0.05); 728 of these events were in the NSR, 443 in the RPE and 120 eOutlier events were found in both tissues. A median number of 3 genes per sample was considered a significant outlier event in the NSR (IQR = 1,4) and 1 in the RPE (IQR = 0,2). In total, we tested 3,209,821 gene-sample events in the NSR and 3,050,081 in the RPE, indicating a significant outlier rate of 0.023% and 0.015%, respectively. These observations are consistent with a recent study of the GTEx cohort, describing significant outlier rates of 0.026%[22].

For each eOutlier event, we were able to harness paired genomic data to identify candidate rare variants potentially driving aberrant expression profiles. We leveraged a hierarchical framework and a probabilistic model to prioritise candidate rare genetic variants driving changes in expression. This identified 230 (23%) eOutlier events likely driven by protein-coding variants and 314 (31%) events with non-coding candidate variants (Supplementary Data 6).

**Rare variants predicted to have a functional impact are identified for 50% of eOutlier events in NSR and RPE.** First, we applied a hierarchical framework to identify rare SVs, CNVs and SNVs which were predicted to result in loss-of-function (pLoF, including frameshift, nonsense and start/stop site loss variants) or were expected to disrupt a nearby non-coding regulatory region

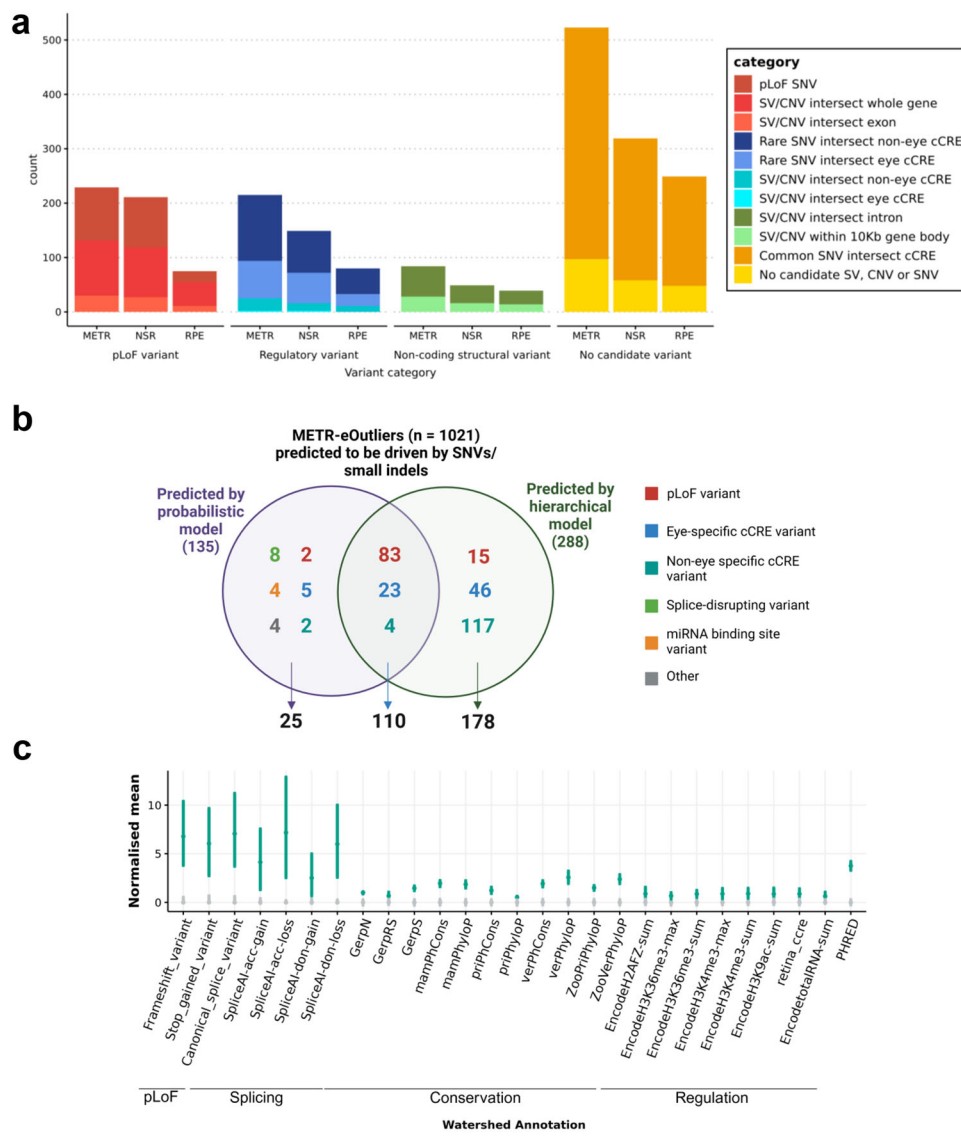

**Fig. 5 | Identification of rare variants driving METR transcriptomic outliers in neurosensory retina (NSR) and retinal pigment epithelium (RPE). a** A hierarchical workflow to identify candidate variants driving outlier expression identified putative functional variants driving 563 eOutlier events in NSR, RPE, or both (METR) **b** The Watershed probabilistic model had high concordance with the hierarchical framework for the prioritisation of SNVs and small indels driving eOutliers in NSR and RPE. **c** Bootstrapping analysis indicates that variants which were predicted to be driving eoutliers by Watershed ($n = 135$) were enriched for pLoF variants, splicing variants, variants within regulatory elements (including retina cCREs) and those with high conservation scores compared to those which were not predicted to have a functional impact ($n = 105,114$). Points indicate median values and bars indicate the 95% CI.

(Supplementary fig. 12). Following this approach, we identified candidate functional variants driving 528 eOutlier events (50.2% of all eOutlier events identified in this study) (Fig. 5A and Supplementary Data 6). Of these, 131 eOutlier events were co-occurring with a SV or CNV impacting the coding-sequence of the outlier gene (77 NSR-only,12 RPE-only and 42 in both tissues) and 98 with a pLoF SNV (77 NSR-only, 6 RPE-only and 15 in both tissues) impacting the same gene (Fig. 5a). For those eOutlier events not explained by an SV, CNV or pLoF SNV disrupting the coding sequence, we identified genomic variants in 71 eOutlier events that were within 10Kb of the gene body and impacted an eye-specific cCRE[18], including SV/CNVs ($n = 2$) and SNVs that are rare (< 0.01 AF) or absent in gnomAD ($n = 69$). We also identified non-eye-specific cCREs from EpiMap[23] which were disrupted by SV/CNVs ($n = 23$) or rare SNVs ($n = 121$) within 10 Kb of the eOutlier gene. Examples of rare variants identified through this analysis strategy are included in Fig. 6 and Supplementary fig. 13.

**A probabilistic model demonstrates high concordance for candidate SNVs driving expression outlier profiles in NSR and RPE.** Next, we applied Watershed[24], a probabilistic model that was retrained on 6 tissue-specific outlier p-values from DROP. This was used in the METR transcriptome datasets to obtain posterior probabilities for SNVs and small indels that may be driving outlier expression profiles. Eye-specific cCREs were added as annotation features for Watershed (Supplementary Data 7) and identified 135 (13%) eOutlier events that were likely to be caused by nearby rare variants (posterior probability > 0.8), of which 110 (81%) were also predicted to be driven by the same variants by the hierarchical model and 11 were predicted to be driven by SV/CNVs that are not considered by Watershed (Fig. 5b). We used bootstrapping analysis to compare the annotations associated with these variants to other rare variants which overlapped with eOutlier genes but were not predicted by Watershed to have a functional impact, observing an enrichment of canonical splice variants, frameshift variants, stop gain variants and variants predicted to disrupt splicing (Fig. 5c). In support

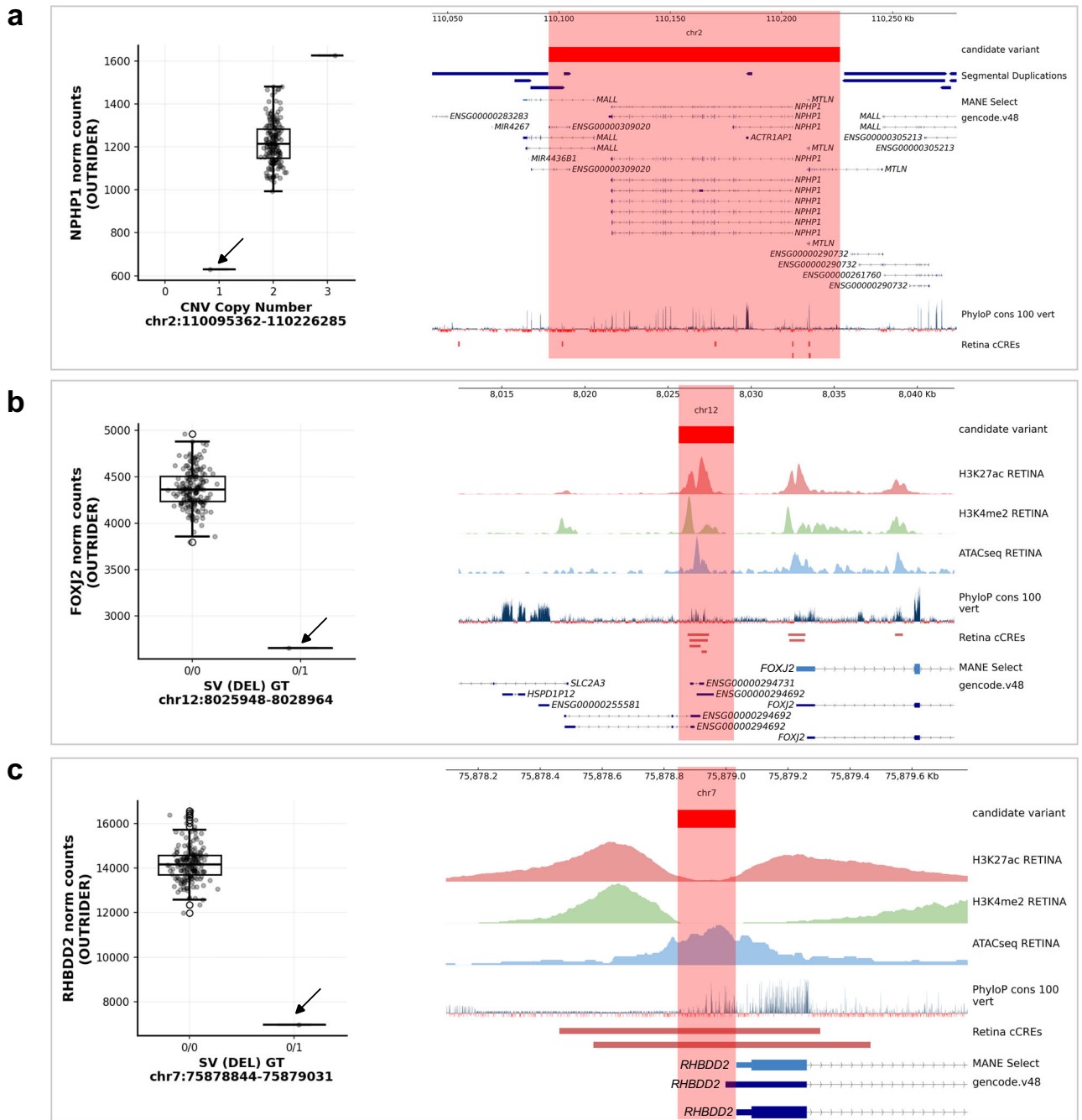

**Fig. 6 | Candidate structural and copy number variants driving METR transcriptomic outliers in neurosensory retina (NSR).** In each caption, the tissue and relative outlier expression profiles calculated through OUTRIDER for individuals are shown, with: **a** copy number states 0 ($n = 0$), 1 ($n = 1$), 2 ($n = 181$) and 3 ($n = 1$); and **b, c** homozygous reference (0/0) ($n = 182$) and heterozygous alternate (0/1) ($n = 1$) genotypes. Genome tracks displaying transcript isoforms, evolutionary conservation and candidate cis-regulatory elements (cCREs) identified in Cherry et al are included alongside: **a** segmental duplication regions; and **b, c** epigenomic histone marks and ATACSeq in retina (Cherry et al.[18]). Box and whisker plots show median values and interquartile ranges, with grey dots indicating normalised count values for single samples and statistical outliers for each genotype indicated with white dots. In **a** CNVs encompassing a known eye disease gene (NPHP1) are shown to cause drastic changes in expression in NSR. In **b, c** deletions are shown to impact cCREs proximal to the transcription start sites of genes expressed in NSR: **b** 3Kb deletion - 3.5Kb upstream of FOXJ2; **c** 187 bp deletion impacting the minimal promoter region of RHBDD2. Box plots show the median (centre line), interquartile range (box; Q1–Q3) and whiskers extending to the most extreme data points within 1.5×IQR. Points beyond the whiskers are plotted as outliers.

of other analyses described in this study, there was an enrichment of rare variants which overlap with retina cCREs and a slight enrichment of rare variants which overlap with epigenomic marks associated with non-eye specific regulatory elements from ENCODE. In total, there were 34 eOutlier events where the functional variants prioritised by Watershed intersected with a known candidate cis-regulatory region (cCRE); 28 of these cCREs were active in the eye.

**Functional assays confirm the impact of rare variants prioritised as drivers of eOutlier expression.** A dual reporter luciferase assay was performed in Human K562 cells to investigate the impact of a *CAND2* heterozygous variant that was prioritised as a driver of drastically reduced expressed in NSR (fold change = 0.6, Z-score = −5.6, p-adj = 0.004) and RPE (fold change = 0.5, Z-score =−4.7, p-adj = 0.049) due to its overlap with features indicative of the *CAND2* promoter region

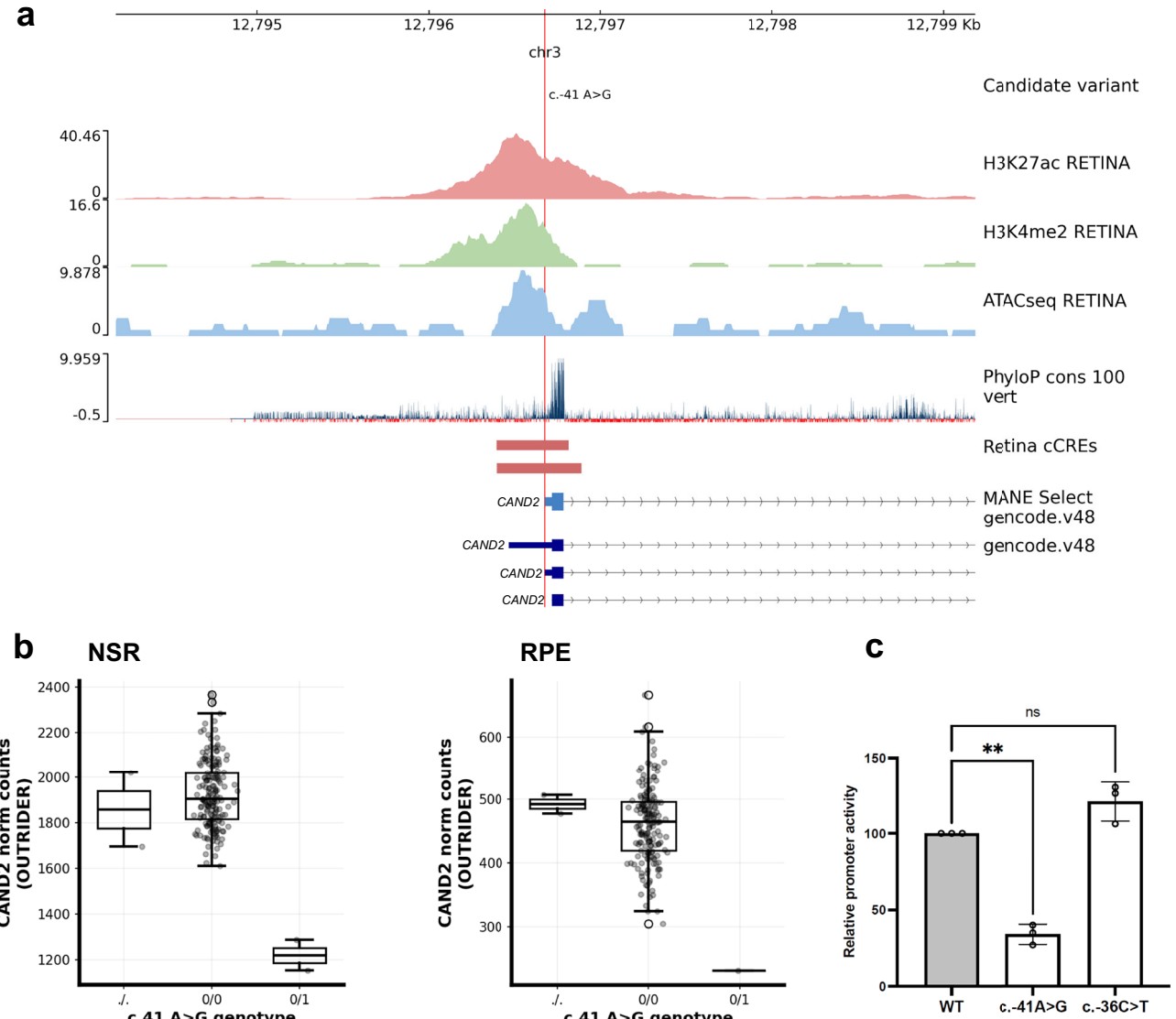

**Fig. 7 | Candidate rare variant in promoter of CAND2 driving METR transcriptomic outliers in neurosensory retina (NSR) and retinal pigment epithelium (RPE). a** Genome tracks displaying the rare promoter variant identified in CAND2 (NM_001162499.2:c.-41A > G) is mapped alongside retina-specific epigenomic peaks and annotated retina-specific cCREs (both generated by Cherry et al.[18], evolutionary conservation and GENCODE v48 transcript isoforms. **b** The outlier expression profiles in NSR and RPE calculated through OUTRIDER for individuals with missing (./.) (*n* = 2 in NSR; *n* = 2 in RPE), homozygous reference (0/0) (*n* = 179 in NSR; *n* = 173 in RPE) and heterozygous alternate (0/1) genotypes (*n* = 2 in NSR; *n* = 1 in RPE) are shown. **c** Results from a dual reporter luciferase assay confirm that there is a significant reduction in CAND2 promoter activity in the presence of c.-41A > G (*n* = 3) (adj *p* = 0.005) compared to the WT promoter (*n* = 3) and a promoter carrying a common variant in gnomAD (c.-36C > T) (*n* = 3). The reported *p*-value was calculated with a one-way ANOVA with multiple comparison test (Dunnett's). Box plots show the median (centre line), interquartile range (box; Q1–Q3) and whiskers extending to the most extreme data points within 1.5× IQR. Points beyond the whiskers are plotted as outliers.

(NM_001162499.2:c.-41A > G; Fig. 7). *CAND2* has recently been implicated in AMD risk through GWAS meta-analysis in European ancestry populations[9] and has an emerging role in the targeted degradation of proteins that is distinct from its *CAND1* homolog[25]. Our dual reporter luciferase assay reveals a significant reduction in *CAND2* promoter activity in the presence of c.-41A > G (adj *p* = 0.005; Fig. 7), confirming the disruption of *CAND2* activity in NSR and RPE and provides a proof-of-principle for our applied prioritisation methods for genetic drivers of expression outliers.

## Discussion

We present a resource to interrogate the impact of both common and rare genomic variation on gene regulation in the human NSR and RPE. We characterised novel eQTL associations that are tissue-specific (Fig. 2) and are enriched to known promoters and proximal enhancers (Fig. 3). We show that eQTLs impacting genes known as a cause of rare genetic eye disease have different properties when compared to those genes which are not known as a cause of eye disease (Fig. 4). We also identify candidate non-coding rare variants, SVs and CNVs which impact cCREs and represent plausible drivers of outlier expression profiles in human NSR and RPE (Fig. 5), including functional validation of a prioritised non-coding genetic variant impacting *CAND2* (Fig. 7). The METR resource can be used alongside other multi-omic datasets to facilitate discovery of novel eye-specific regulatory elements, including those implicated in common (e.g. AMD) and rare (e.g. IRDs) genetic disorders impacting the retina.

The cohort of 201 human donors described in this study represents the first dataset, to our knowledge, to pair whole genome

sequencing with high-depth RNA sequencing data from the NSR and RPE. Previous studies have developed RNA sequencing from the NSR alongside genotyping arrays[15,16,26] and this has enabled the characterisation of eQTLs in the retina including preliminary data supporting the role of a limited number of eQTLs in AMD (Table 1). We performed extensive QC for both DNA and RNA sequencing data to confirm the validity of the datasets generated in this study, in particular due to the prolonged median ischemic time compared to GTEx samples, which may impact the quality of the data obtained[27]. These analyses confirmed suitable RNA integrity values across the cohort (Supplementary fig. 1), high unique mapping rates with appropriate read lengths and appropriate 3′/5′ biases from RNAseq data (Supplementary figs. 1 and 4), along with representative gene expression profiles (Fig. 1c and Supplementary Data 4). The high-depth and high-quality RNA and whole genome sequencing datasets developed for this study are from a cohort of individuals without clear signs of late-stage AMD and have enabled biological insights beyond those described previously.

Firstly, we were able to assess whether previously characterised eQTLs are replicated utilising alternative methods and technologies in an independent cohort of individuals of European genetic ancestry without late-stage AMD (Supplementary fig. 3). As gene expression profiles have been shown to be significantly disrupted during AMD pathogenesis[28,29], it is important to identify eQTL signals that are amplified or disrupted by broader changes in transcriptome profiles associated with AMD, as well as those that remain consistent within a cohort of individuals without clear signs of late-stage AMD. Overall, we show high levels of replication of eQTL findings from Ratnapriya et al., with 5993 identical eGenes and replication of 13 eQTLs previously implicated with a role in AMD (Table 1), including *PILRB,* which has recently been shown to lead to photoreceptor dysfunction in mice when function is impaired[30]. Notably, we identified 5 novel QTLs for genes previously implicated in AMD (*ACAD10, HTRA1, B3GLCT, PLA2G12A, BAIAP2L2*) and 4 genes implicated in AMD without replication of a previously characterised eQTL (*CFI, COL4A3, RDH5, TNFRSF10A*). This suggests that differences in the approaches undertaken and/or cohort composition, e.g. AMD status, cohort size and/or genetic ancestry, impact the influence of genomic drivers on the expression of these genes.

Second, variants which are rare in the population or unique to individuals have been demonstrated to drive drastic changes in expression profiles, so-called 'expression outliers', across different tissues[24,31]. The use of complete genomic sequencing in this cohort, achieving a median coverage of 36x, has enabled the characterisation of a greater diversity of genomic variation than has previously been studied in the context of expression drivers in the NSR and RPE and identified thousands of new regions which can be interrogated for rare variation within disease cohorts[32]. Using two distinct variant prioritisation approaches, we describe rare variants in the general population, including SVs, CNVs and small variants that are the most likely drivers of expression outliers in these tissues (Fig. 5). Through functional validation of a prioritised non-coding variant in the *CAND2* promoter region, we establish a proof-of-principle for the applied variant prioritisation approaches (Fig. 7) and provide mechanistic insight into non-coding regions regulating the expression level of AMD-risk associated genes[9]. These data encourage further functional follow-up for the 578 prioritised variants that may be causative of pronounced changes in expression profiles in the human retina, including 272 rare variants predicted to cause loss-of-function and 299 that intersect with non-coding regions (including examples presented in Fig. 6 and Supplementary fig. 13). Other recent studies have identified outlier-associated non-coding rare variants that contribute to common disease predisposition[33] and underpin rare genetic disorders[34,35]. Moreover, non-coding variation has been identified as a cause of genetic ophthalmic disorders in untranslated regions[36], retina-specific exons[37], promoters[38], distal enhancers[39] and non-coding

genes[40] expressed in the NSR and RPE. With the increasing availability of genomic sequencing datasets for the diagnosis and discovery of genetic disorders[41,42], including ophthalmic conditions[43], the data developed in this study is timely and provides an opportunity, alongside other complementary datasets[44], to identify new pathogenic mechanisms underpinning genetic disorders.

Third, we have generated high-coverage RNAseq datasets achieving, on average, 139 million uniquely mapping reads for NSR and 62 million uniquely mapping reads for RPE. Previous studies have developed lower coverage RNAseq datasets for NSR, for example, EyeGEx[15], Orozco et al. [26] and Pinelli et al.[45] generated 33, 30 and 72 million sequencing reads per sample, respectively. Previous studies have remarked on the level of transcript diversity in NSR[46] and highlighted the advantage of high-depth RNAseq in this context. In comparison to EyeGEx, our high coverage approach elevates the number of observable protein-coding genes by 23% (from 13,662 to 16,765) and newly identifies 3,663 eGenes. For 4,481 eGenes that are not replicated from EyeGEx, further study, for example harmonisation of genomic and RNA sequencing dataset processing and meta-analyses would assist in understanding whether their detection is influenced by cohort composition, methodologies undertaken and/or sample size. The increased number of eGenes from this study enables observation of patterns in gene expression at increased resolution and has granted insight into the trends associated with genes previously implicated in genetic disorders impacting vision. Overall, we observed that eGenes that have been characterised as a cause of rare genetic eye disease[20] have lower expression variability across individuals than non-disease genes (Fig. 4), suggesting that regulation of these genes is more tightly controlled in NSR and RPE. The role of eQTLs in genetic disorders remains incompletely understood. For example, whilst some studies have shown eQTLs contribute to onset, penetrance and expressivity[47,48], including genetic disorders impacting the eye[13], others have found limited evidence for their role in neuronal genetic disorders[49,50]. Here, we observe that eQTL variants which were associated with changes in expression of eye-disease genes had significantly lower effect sizes and their allele frequency was higher than eQTLs impacting genes that have not previously been implicated in eye disease (Fig. 4 and Supplementary fig. 11). Intuitively, the absence of rarer and higher impact eVariants amongst a population of individuals without signs of genetic eye disease suggests constraint on genomic variation with these properties, although population-scale modelling and statistical analysis is required to formally test this hypothesis.

Finally, as our cohort includes 158 individuals with RNA extracted and sequenced from both NSR and RPE datasets, this enables further insights into the expression patterns and regulatory architecture of these tissues, unbiased by sample preparation methods and/or differences between individuals, e.g. genomic background. It should be noted that our cohort is biased towards male individuals (64%) and this may have a hidden bias on eQTLs and transcriptome differences identified. However, our data newly identifies 916 eGenes in NSR and RPE compared to those characterised in other tissues[10] and we observe a high level of overlap in eGenes between NSR and RPE, including 86% of RPE eGenes and 32% of NSR eGenes. These data further support the high level of overlap previously observed for active enhancers and promoters between RPE/choroid and NSR[18].

Whilst the findings of this study have enhanced our understanding of genomic regulation in human NSR and RPE, other approaches that utilise single-cell[26,51–54] single-nuclei[17,55] and spatial[56,57] transcriptomic approaches enable increased precision to understand gene expression in specialised retinal layers and cell types. These approaches are particularly advantageous for the NSR, which is a highly heterogeneous tissue comprised of several specialised layers and neuronal cell types, including photoreceptors, bipolar cells, amacrine cells and horizontal cells[58] and where transcriptome profiles may differ substantially between the central and peripheral retina[59] To

overcome potential shortcomings of the bulk RNAseq approach adopted in this study, we performed deconvolution analyses to estimate the relative sample composition against single-nuclei RNA-sequencing[17]. Given the complexity associated with retinal tissue dissection and storage[60,61], the deconvolution approach also enabled confirmation of tissue sample integrity alongside differential expression profiles (Fig. 1c). Bulk RNAseq from NSR had representation, as expected, from diverse cell types with significant enrichment towards rod photoreceptors and retinal astrocytes, representing >50% of the estimated cellular make-up of most samples. In keeping with current understanding of retinal ageing[62] there is observed a significant loss of rod photoreceptors with age (Supplementary Fig. 5B). However, deconvolution is naturally limited by the relative differential transcriptional activity between cell types and is complicated by cell types with similar transcription profiles, for example, between Müller glia and retinal astrocytes[53]. We expect that the high number of astrocytes predicted in RNA samples is influenced by similar transcription profiles to other cell types and whilst we confirm that we have generated high-quality RNA sequencing datasets (Supplementary figs. 1 and 4 and Fig. 1), these estimates may also be influenced by altered transcriptome profiles in samples with longer ischemic times[27] and/or the high sequencing depth coverage generated. Moreover, the retina is known to have cyclic patterns of gene expression, related to both circadian rhythm and natural function, i.e. response to light[63] and as such, there is an incomplete molecular understanding of all cell types present in the human retina[17]. Overall, these data support the integrity of the RNAseq dataset developed in this study and whilst confident quantification of the cell types present is not possible, our analyses confirm that the datasets are representative of major cell types in the retina.

Taken together, the data presented in this study provide new insights into the genomic control of gene regulation in the human retina. We build upon previous understanding through replication of eQTLs in a cohort of individuals without clear signs of late-stage AMD, characterise hundreds of new genes under genomic regulation and provide insights into the role of rare variants, SVs and CNVs in the disruption of gene expression in these specialised tissues that enable vision. Future studies utilising this resource, including meta-analysis with other published datasets, co-localisation and transcriptome-wide association studies incorporating findings from genome-wide association studies, will continue to develop understanding of the expression profiles and the role of non-coding genetic variation in the onset and presentation of genetic disorders impacting vision.

## Methods

### Ethics approval and material transfer
All research and approaches undertaken in this manuscript were approved by the North West−Greater Manchester Central Research Ethics Committee and NHS Health Research Authority (15/NW/0932). Methodological approaches were approved and undertaken at The University of Manchester. The Manchester Eye Tissue Repository is a non-profit tissue bank, and no compensation was provided for the receipt or delivery of tissue samples. A material transfer agreement was agreed upon between the research team and the Manchester Eye Tissue Repository. Any surplus tissue, RNA and/or DNA after sample preparation and sequencing remained with the research team and will be destroyed or returned to the tissue bank within 3 years of the conclusion of the study. Other researchers who wish to access surplus material can submit independent requests to the Manchester Eye Tissue Repository.

### Gene expression quantification in neurosensory retina and retinal pigment epithelium from RNA-Seq data
Paired-end short-read sequencing of polyA-enriched mRNA (RNAseq) was performed on an Illumina NovaSeq 6000 instrument for two layers of the retina: (1) the entire neurosensory retina (NSR), including macula and peripheral regions and (2) pelleted cells from the retinal pigment epithelium (RPE), which were scraped from Bruch's membrane. Donor eye tissues were obtained from the Manchester Eye Tissue Repository, an ethically approved Research Tissue Bank (UK NHS Health Research Authority, 15/NW/0932). Eye tissue was acquired after the corneas had been removed for transplantation and explicit written informed consent had been obtained from donors or their next of kin to use the remaining tissue for research. Samples were selected for RNAseq with reference to RNA concentration (ng/μl) and integrity (RNA Integrity Number−RIN) values, calculated with the Agilent TapeStation system. The QC process was performed without knowledge of sample sex/gender. (see Supplementary Methods 1.1 and 1.2 for additional details on tissue extraction and the RNA sequencing protocol).

The Genotype-Tissue Expression (GTEx) analysis pipeline[64] was applied to RNAseq datasets to assess quality and to perform alignment and expression quantification. Alignment was performed against the GRCh38 human reference genome using STAR v2.7.4a[65]. Duplicate reads were marked with Picard v2.27.1[66]. Gene-level expression quantification, using the GENCODE v38 annotation[67] was carried out using RNA-SeQC 1.1.9[68], for gene-level read counts and RSEM v1.3.0[69], for gene-level quantifications in transcripts per million (TPM). Quality assessments of processed RNAseq datasets included reference to the total number of reads, number of uniquely mapped reads, number of splice junctions, number of chimeric reads, read length and 3'/5' bias for all NSR and RPE samples. To ensure concordance between paired WGS and RNAseq samples, we excluded WGS-RNAseq pairs where the predicted relatedness, calculated using *Somalier*[70], was <0.8.

### Whole genome sequencing data
Short-read paired-end whole genome sequencing (WGS) was generated for each donor on an Illumina NovaSeq6000 Instrument using DNA extracted from iris biopsies (see Supplementary Methods Section 2 for additional details). Genome alignment and variant calling were carried out using Illumina DRAGEN 4.0.3 software with Machine Learning and Graph Map Enabled. Aggregate variant detection and harmonisation were carried out using Illumina DRAGEN 4.0.3 software Population Mode. We applied quality control filters to the aggregate VCF to remove low-quality variant calls using a combination of *bcftools* (v.1.16) and *PLINK* (v.2.0) (see Supplementary Methods 2.3). For eQTL mapping, aggregate genotypes were binarized using PLINK 2.0.

### Cell type deconvolution of bulk RNA-seq data
We used BayesPrism (Bayesian cell Proportion Reconstruction Inferred using Statistical Marginalization)[71] to run a deconvolution model to estimate the proportion of retinal cell types in the generated bulk RNA-seq data in NSR and RPE. The reference dataset to train the model was a single-cell RNAseq dataset from the ocular posterior segment[17] (See Supplementary Methods 1.5 for additional details).

### Differential expression analysis between NSR and RPE
To ensure the validity of the transcriptomic datasets generated in this study, we assessed the biological relevance of expressed genes in NSR and RPE. We used the R package *deseq2*[72] to identify genes that were differentially expressed between NSR and RPE. We included age and sex as covariates in the *deseq2* model with the false discovery rate threshold set at 0.05. To confirm *deseq2* results, we replicated the differential expression analysis using *edgeR*[73].

To identify which gene ontology biological pathways were enriched in the upregulated genes in NSR/RPE, we carried out gene set enrichment analysis (GSEA) of the genes that were differentially expressed between both tissues (FDR < 0.05), using *WebGestalt*[74]. We processed the GSEA output with a clustering algorithm, *rrvgo*[75], to

group similar GO terms together and selected representative terms. (See Supplementary Methods 1.6 for additional details).

### Input Data for cis-eQTL analysis

For eQTL analysis, we generated a normalised expression matrix for each tissue. Genes that did not meet expression thresholds of >0.1 TPM in at least 20% of samples and ≥6 reads in at least 20% of samples were removed from eQTL analysis. Expression values were normalised using the trimmed mean of M-values normalisation (TMM) method[76] and using an inverse normal transform.

To account for known and unknown biological and experimental confounding factors, a set of 30 covariates was generated for each RNA-Seq sample using the Probabilistic Estimation of Expression Residuals (PEER) method[77] applied to normalised gene expression levels.

Principal component analysis with EIGENSOFT 6.0.1[78] was carried out to capture ancestral variation within the cohort. The top five principal components for each participant were used as covariates in the eQTL analysis.

### Cis-eQTL mapping with tensorQTL

TensorQTL[79] was used to identify genetic variants that were significantly associated with the expression of nearby genes (up to 1 Mb away) in NSR and RPE (FDR < 0.05). The required input files were the normalised gene expression matrix, the binary and filtered genotype data and a covariates table which included the following information for each participant: sex, WGS batch, five top principal components and 30 PEER factors. To quantify the eQTL effect size, we estimated the log2 allelic fold change (aFC), following the method established by Mohammadi et al.[80] (see Supplementary Methods section 4 for additional details).

### Comparison with other eQTL studies

We compared all METR eQTLs with retina eQTLs mapped by EyeGEx[15] and Strunz et al.[16]. We identified genes had been associated with eQTLs in our study and in EyeGEx and/or Strunz et al. (eGenes shared between studies). For these shared eGenes, we extracted the top eQTLs identified by EyeGEx and/or Strunz et al. and checked if they were replicated in our cohort or if they were in high LD (r2 > 0.8) with a METR-NSR eQTL. Pairwise LD scores were calculated using LDlinkR[81].

To compare to non-retina tissues, all significant eQTL associations were downloaded from the GTEx Open Access portal (v8) for each available tissue (https://www.gtexportal.org/home/downloads/adult-gtex/qtl). We calculated the intersection between the number of METR-eQTLs and eGenes which were also shared by each GTEx tissue using the Intersection over Union (IoU) statistic. The IoU calculates the ratio of the number of eQTLs/eGenes present in both sets over the total number of eQTLs/eGenes in one set and/or the other.

### Annotation of eVariants and bootstrapping analysis to calculate enrichment of eQTLs in characterised regulatory loci

All NSR and RPE eQTL variants were annotated with the Ensembl Variant Effect Predictor[82]. We assessed overlap and annotated all eVariants with a set of tissue-specific and cell-type-specific annotations of candidate cis-regulatory elements (cCREs) from a variety of sources (Supplementary Methods Table 1). These included characterised regulatory loci from retina, RPE and macula[18], cell-type specific regions of open chromatin detected by scATACseq from retina samples[19], non-eye specific cCREs from adult tissues in EpiMap[23] and cell-type agnostic candidate cis-regulatory element (cCRE) annotations from ENCODE[83].

To calculate the relative enrichment of eVariants that overlapped with each type of regulatory element, we used bootstrapping analysis. We carried out 1000 iterations by subsampling 100,000 random eVariants with replacement and 100,000 control variants from our cohort that were included as input for the eQTL mapping and did not

meet the eQTL significance threshold (FDR < 0.05), matched for gene density and allele frequency. We compared the ratio of eVariants to control variants that intersected with each type of regulatory element. (see Supplementary Methods section 5 for additional details).

### Analysis of the properties of eQTLs that impact known eye disease-related genes

To understand if there were trends that were specific to eQTLs associated with known monogenic eye disease genes, we utilised the EyeG2P resource[20]. All other METR-eGenes were considered non-eye disease genes. We compared the eQTL/eGene properties between eye-disease and non-eye-disease gene eQTLs, including eVariant allele frequency and effect size, measured using log2 allele fold change. (See Supplementary Methods section 6 for additional details.

### Identification of transcriptome outliers using the DROP pipeline

We utilised the DROP v.1.4.0 pipeline[21] to identify transcriptome outliers from NSR and RPE, using standard parameters.

### Hierarchical workflow to identify candidate variants driving outlier expression

We developed a hierarchical workflow to identify candidate variants driving outlier expression (eOutliers) using snakemake version 7.32 (Supplementary Fig. 14). Briefly, the workflow would first search for a pLoF variant from the eOutlier sample in the corresponding eOutlier gene, which could be an exonic structural variant, or a SNV with a high impact consequence based on *Ensembl's Variant Effect Predictor (v.112.0)*. If no pLoF variant could be identified, the workflow would then search for regulatory variants that were within 10Kb of the eOutlier gene body. Regulatory variants were defined as structural variants and rare SNVs (gnomAD AF<0.01) that overlapped with nearby retina cCREs or non-retina-specific cCREs from different adult tissues in EpiMap. If no regulatory variant was identified, the model would identify any other non-coding structural variant within 10Kb of the eOutlier gene body, before returning a negative search result (see Supplementary Methods section 8 for additional details).

### Implementation of the watershed

For all genes with an eOutlier in the NSR, we extracted all rare variants (gnomAD allele frequency <1%) that intersected with the gene body ± 10Kb. Variants were extracted for all samples with NSR RNAseq data from the post-QC aggregate VCF. We annotated all rare variants with selected annotations from VEP[82] and CADD[84] (Supplementary Data 6) and intersected them with known retina-specific cCREs from Cherry et al.[18] and non-retina-specific cCREs from EpiMap. Missing annotations were replaced with default imputation values obtained from CADD (Supplementary Data 6). The Watershed model[24] was run using the predict_watershed.R script with an adjusted *p*-val threshold of 0.05 and the number of dimensions set to 6. (See Supplementary Methods section 9 for additional details).

### Dual reporter luciferase assay

A 294 bp fragment of the wild-type promoter region from *CAND2* was PCR-amplified from control genomic DNA using Phusion High-Fidelity DNA Polymerase (Promega). To introduce variants, two overlapping fragments were amplified using a combination of mutagenic primers. Variants constructed were the variant of interest, NM_001162499.2:c.-41A > G and a variant that is common in the general population and not expected to impact *CAND2* expression, NM_001162499.2:c.-36C > T.

The wild-type and variant fragments were assembled into NheI-NcoI digested pGL4.10[luc2] firefly luciferase plasmid using the Gibson method. The assemblies were transformed into competent E. coli grown overnight on LB agar containing carbenicillin. Candidate colonies were picked for culture and plasmid isolation. The plasmid constructs were verified by Sanger sequencing. Human K562 cells were

transiently transfected with 500 ng of plasmid using Lipofectamine LTX (Invitrogen) following the manufacturer's standard protocol. An empty pGL4.10[luc2] plasmid was transfected as a control for background activity. The Renilla luciferase pGL4.74[hRluc/TK] vector (Promega) was co-transfected as an internal luminescence control. Following 20–24 hr incubation at 37 °C with 5% $CO_2$, a dual luciferase assay was conducted using the Dual-Glo® Luciferase Assay (Promega).

## Reporting summary

Further information on research design is available in the Nature Portfolio Reporting Summary linked to this article.

## Data availability

All raw RNA sequencing and genomic sequencing datasets generated in this study are available under controlled access through the European Genome-phenome Archive (EGA; Study ID: EGAS50000001443; Dataset: EGAD50000002082). Processed datasets, including eQTL results, eOutlier statistics and aggregated genomic variant files, are available under controlled access through the EGA (Study ID: EGAS50000001443; Dataset: EGAD50000002082). Controlled access to these datasets is a condition of access to tissue samples from the Manchester Eye Tissue Repository to ensure traceability of data access and usage, as per conditions of ethical approval for the biobank (15/NW/0932). Applications for access to the raw and processed datasets can be made through the EGA and will receive a response by the EGAC50000000807 EGA Data Access Committee within 4 weeks of the data access request. The full data access policy and terms of data usage are available through the EGA. The Genotype-Tissue Expression (GTEx) Project[10] data used in this study are available from the GTEX public portal at https://www.gtexportal.org/home/. The ENCODE Project Candidate cis-Regulatory Element Registry V3[83] data used in this study are available from the SCREEN portal at https://screen.encodeproject.org/. The Epigenome Integration across Multiple Annotation Projects[23] (EpiMap) data used in this study are available from the EpiMap Repository at https://compbio.mit.edu/epimap/. The retina-specific epigenomic tracks generated by Cherry et al.[18] used in this study are available as custom tracks on the UCSC browser and were accessed from https://tinyurl.com/CherryLab-EyeBrowser. The single-cell RNAseq data from the ocular posterior segment generated by Monavarfeshani et al.[17] used in this study are available at the Broad Institute Single Cell Portal, under study number SCP2310, accessible from https://singlecell.broadinstitute.org/single_cell/. The retina scATACseq peaks generated by Wang et al.[19] used in this study are available in the GEO database under accession code GSE196235. The EyeGex[15] used in this study is available under controlled access; access was obtained by contacting the corresponding author of the study. The retina and RPE-specific eQTLs generated by Orozco et al.[26] used in this study were accessed from https://eye-eqtl.com/ in April 2024. The retina-specific eQTLs generated by Strunz et al.[16] used in this study are publicly available and were accessed from http://www-huge.uni-regensburg.de/ in October 2025. The 1000 Genomes Project V3 data used in this study are publicly available from the 1000 Genomes Project Public Portal at https://www.internationalgenome.org/data/. The Genome Aggregation Database (gnomAD v4) data used in this study are publicly available from the gnomAD Public Portal at https://gnomad.broadinstitute.org/data. All other data supporting the findings of this study are available in the article and its Supplementary Information files.

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

## Acknowledgements

We express our sincere thanks to the donors and their families for enabling this research. We thank Selina Mcharg, Nadhim Bayatti and Jay Brown for the development of the Manchester Eye Tissue Resource. We also thank staff at the University of Manchester Genomic Technologies Core Facility and at the Ocular Genomics Institute, Harvard Medical School, for their help in the generation of DNA and RNA sequencing datasets for this study. The views expressed are those of the authors and not necessarily those of the funders, including the NIHR and the Department of Health and Social Care. J.E discloses support for the research of this work from the Macular Society (United Kingdom), Fight For Sight, the UK Medical Research Council and the NIHR Manchester Biomedical Research Centre (NIHR203308). J.S discloses support for the research of this work from the UK Medical Research Council (MR/W007428/1). P.I.S discloses support for the publication of this work from the Wellcome Trust (224643/Z/21/Z, Clinical Research Career Development Fellowship) and the UK National Institute for Health Research (NIHR) Clinical Lecturer Programme (CL-201-06-001). D.B discloses support for the publication of this work from the NIHR Research Professorship grant (RP-2016-07- 011). A.V.S discloses support for publication of this work from the NIH/ NEI (R01 EY031424). K.M.B discloses support for publication of this work from the NIH/NEI (R01EY035717).

## Author contributions

J.M.E., A.V.S., K.M.B., P.I.S., D.B. and G.C.B. conceived the study and obtained funding. J.S. led and performed the analysis described in the study under the supervision of J.M.E. and S.B. J.A.-D. performed cellular deconvolution analyses under the supervision of J.M.E. S.J.C. and P.N.B. collected retinal tissue and performed quality and age-related macular degeneration assessments. SH1, BA, SH2 and AH extracted molecular material and generated the paired DNA and RNA sequencing datasets. R.A.K., H.B.T. and R.T.O'K. designed and performed dual reporter luciferase assays. J.S. and J.M.E. wrote the manuscript with review and critical input from all authors.

## Competing interests

The authors declare no competing interests.
