## [Transparent Peer Review file · Nature Communications]

Paired DNA and RNA sequencing uncovers common and rare variation regulating human retinal gene expression

Corresponding Author: Dr Jamie Ellingford

Version 0:

Reviewer comments:

Reviewer #1

(Remarks to the Author)

The manuscript by Sampson, et al. utilizes whole genome sequencing and RNA-Seq across ~201 human donor eyes to better understand gene expression regulation in the neural retina and RPE. They identify nearly 1.5 million eQTLs that affect over 13,000 genes in the retina and RPE. Further, they identify expression outliers that can be attributed to SNPs, structural and copy number variants. The authors attempt to make this data translatable to ocular disease, in this case, specifically to AMD. Overall, the research strategy is sound, however, there are major and minor concerns that need to be addressed.

Major concerns

1. Average PMI of 40 hours likely introduces a lot of artifacts in the RNA-Seq data. This needs to be addressed in the Discussion.
2. Given the high PMI, the average RIN should be given for the samples. They are likely below 7, which is the baseline for RNA-Seq. This does not mean the samples are invalid, but high 5' bias will likely affect the results.
3. Similarly, using 25 ng to 1 µg of RNA for library preparation is not a good practice. This is because the samples will require significantly different number of PCR cycles. The lower yield samples will have more duplicates and less representation of lower expressing genes.
4. Based on the Methods, it is not clear that duplicate reads were removed from the data prior to quantification.
5. The differential expression algorithm used should be listed in the methods.
6. It's mentioned that 49 eyes have AMD risk SNPs, but no pathology. Have other common diseases been ruled out (glaucoma, diabetic retinopathy, etc.)?
7. Since such a large focus was placed on translation to AMD, it seems that neither the macula RPE, or retina were sampled. It's been shown that the macula RPE differs transcriptionally from the peripheral RPE. And no cones were among the list of deconvoluted cell type of the retina. This makes the AMD association difficult.

Minor comments

1. It's unclear why IRDs are mentioned in the Introduction. This leads the reader to think they'll be discussed with the data but are not.
2. The authors mention that higher depth of sequencing was achieved in the current study compared to those in the literature. In all studies, the numbers generated for the RPE likely reach 90% or greater depth of coverage. If the authors would like to keep that statement in the Discussion, then they should provide depth of coverage plots for their samples.
3. The reporting summary states that the data will be available through dbGAP, yet this is not mentioned in the manuscript.

Reviewer #2

(Remarks to the Author)

Genetic variation influencing retinal gene expression is still not fully understood, particularly in the context of vision-related disorders. In this study, the authors performed whole-genome and RNA sequencing on neurosensory retina (NSR) and retinal pigment epithelium (RPE) samples from 201 post-mortem eyes. They identified over 1.4 million significant cis-eQTLs as well as 299 rare variants that may contribute to expression outliers. These findings advance our knowledge of both common and rare genetic regulation in human retinal tissues. This reviewer has the following comments for the authors to

consider:

1. Numerous studies, including GTEx, have consistently demonstrated that the discovery power for eQTLs increases substantially with larger sample sizes. The authors correctly note that a large-scale eQTL study in the retina has already been published. However, in the present study, they choose to analyze their dataset of 200 retina samples independently and only compare their results with the previously published dataset. This approach represents a missed opportunity. By combining their data with the existing dataset, the authors could have achieved a significantly larger and more statistically powerful sample size, potentially uncovering additional and more robust eQTLs. I strongly recommend that the authors consider performing a joint or meta-analysis with the previously published data. Such an integrative approach would enhance the impact and biological insight of the study.
2. The comparison of METR-NSR eQTLs for each eGene identified in this study with the retina-specific eQTL dataset from the EyeGEx project is not entirely appropriate. The EyeGEx study focused exclusively on gene expression in the neural retina, whereas the METR-NSR dataset includes eQTLs derived from both the neural retina (NSR) and the retinal pigment epithelium (RPE), either separately or in combination. As a result, the eQTLs identified in METR-NSR may reflect regulatory variation specific to the RPE, which would not be captured in a retina-only dataset like EyeGEx. To ensure a more accurate and biologically meaningful comparison, the authors should separate the RPE-derived eQTLs from those identified in the NSR and restrict their comparison with EyeGEx to the NSR component. This would allow for a more valid one-to-one comparison and avoid confounding results due to tissue-specific regulatory differences.
3. In this study, the authors have analyzed cell-type agnostic cis-candidate regulatory elements (cCREs) as defined by ENCODE (v3). While ENCODE provides a valuable and comprehensive resource, it is based on broadly defined regulatory elements that may not capture the specificity of gene regulation in ocular tissues. Importantly, several high-quality studies have been published that provide retina- and RPE-specific chromatin accessibility and regulatory element datasets, including those by Wang et al. (2022), Orozco et al. (2020), Wang et al. (2023), and Cherry et al. (2020). These resources offer cell-type or tissue-specific regulatory insights that are more biologically relevant to the context of this study. I strongly recommend that the authors incorporate these eye-specific datasets into their analysis to derive more nuanced and biologically meaningful interpretations. Doing so could enhance the resolution and relevance of their regulatory annotations, particularly in understanding tissue-specific gene regulation mechanisms in the retina and RPE.
4. While the use of whole-genome sequencing (WGS) data to investigate the role of rare variants in gene expression regulation is innovative and represents a valuable direction for the field, the current study appears significantly underpowered for such an analysis. The sample size is relatively small for detecting rare variant effects, which typically require much larger cohorts to achieve sufficient statistical power. As reported, only 1,051 expression outliers were identified, and among these, just 299 rare non-coding single-nucleotide, structural variants or copy number variants as plausible for 28% of outlier events were prioritized as plausible contributors—accounting for approximately 28% of the outlier events. While this is a noteworthy effort, the modest number of supported associations limits the strength of the conclusions that can be drawn. As a result, the overall enthusiasm for the utility of this dataset to robustly explore rare variant-driven regulatory mechanisms remains tempered.
5. The authors state that the METR resource can be integrated with other multi-omic datasets to support the discovery of novel eye-specific regulatory elements, including those potentially involved in both common (e.g., AMD) and rare (e.g., IRDs) genetic disorders affecting the retina. While this is an important and promising application of the resource, the current manuscript does not present any direct evidence or in-depth analysis specifically demonstrating the utility and power of METR for studying rare variant contributions to IRDs. Given the complexity and low allele frequencies typically associated with rare disease-causing variants, more focused analyses or case studies—linking METR data with known IRD loci or variant annotations—would help substantiate this claim. As it stands, the utility of METR for rare variant interpretation remains speculative. Figure 6 and 7 present selected examples, but doesn't not provide much functional relevance. Given the challenges associated with interpreting the regulatory impact of rare, non-coding variants, functional follow-up is essential to establish biological relevance. Including such validation—whether through reporter assays, CRISPR perturbations, or integration with existing disease-linked variant databases—would greatly strengthen the authors' claims and enhance the translational potential of the METR resource for rare disease research.
6. The authors claim that the higher sequencing depth used in their study leads to improved gene detection and expression quantification. Specifically, they report that compared to EyeGEx, their high-coverage approach increases the number of detectable protein-coding genes by 23% (from 13,662 to 16,765) and newly identifies 3,663 eGenes. It would be important to clarify whether these gene counts and newly identified eGenes are derived exclusively from neural retina (NSR) samples, or whether they also include data from the retinal pigment epithelium (RPE), which could confound direct comparisons with EyeGEx, a retina-only dataset. Additionally, there appears to be a discrepancy in the reported overlap with EyeGEx findings. EyeGEx identified approximately 10,000 eGenes, but, yet the current study reports overlap with only 5,993 NSR eGenes from EyeGEx. The authors should address this gap and provide an explanation as to why a substantial number of previously reported eGenes were not detected in their dataset. Potential contributing factors—such as differences in sample size, statistical power, expression thresholds, or tissue composition—should be discussed to help contextualize this observation.

Minor points

1. Page 12, lines 288–293, contains a repetitive statement about sequencing depth that was already addressed in the second paragraph of the Discussion. The authors may consider removing or consolidating this section to avoid redundancy.
2. The manuscript currently lacks a statement regarding data availability. Could the authors please include information on how and in what form the data will be shared, including any relevant accession numbers or repositories?

Reviewer #3

(Remarks to the Author)

This manuscript describes analyses of whole genome sequencing and bulk RNA-sequencing of neurosensory retina (NSR) and retinal pigment epithelium (RPE) from 201 post-mortem eyes. The authors have reported differential expression analysis of genes (NSR vs RPE), identified eQTLs for NSR and RPE, and compared these to published retina eQTLs, GTEx eQTLs and eGenes. The authors have also integrated cis-regulatory elements (cCREs) from ENCODE and detected expression outliers and prioritized 299 rare non-coding single-nucleotide, structural variants or copy number variants.

Overall, the authors have attempted to put together a good resource for retina and RPE transcriptome and eQTLs. However, there are numerous issues pertaining to analyses and significance. If the analyses are performed using similar parameters, it is unclear how this study would compare with what has already been published with larger study samples (see comments below).

Major issues:

- (i) Small number of retina samples (about 200) compared to Ratnapriya et al. Nat Genet 2019 which included 453 controls and AMD cases. Another paper Strunz et al. PLoS Genet 2020 that was not included by the authors used 311 healthy human retina samples for eQTL analysis. Thus, the study is underpowered for eQTL analyses compared to published studies, at least for the retina.
- (ii) RPE samples have greater variability (line 67) because of biological and technical reasons. RNA from older RPE samples is generally degraded. The authors have not provided any information with respect of sample collection, RNA quality and other variables, which can significantly impact the expression variability. It's unclear if this data is trustworthy.
- (iii) Both Ratnapriya and Strunz studies used more conservative expression values of CPM>1 or >2 in 10 or 50% of samples. This study appears to have selected TPM values of >0.1 for subsequent analyses resulting in 28,512 genes. This is a rather high number. It's unclear if a change in TPM of 0.1 to 0.2 or 0.5 (which are 2-5-fold change) even if statistically significant would have any biological relevance. At the very least, the authors need to repeat the complete eQTL analysis with TPM of >1 in 10-20% of samples and with lesser number of genes.
- (iv) The authors have identified 21,157 differentially expressed genes with adjusted p value of 0.01 between NSR and RPE. The number is too high and unreasonable. The significance of this differential expression analysis between neural retina and RPE is unclear. These are two completely different tissues, and such large differences in gene expression do not add much value. Common genes will likely be constitutively expressed genes. Even otherwise, how was the differential expression analysis done (the methods are missing)?
- (v) The simple reason for disease and non-disease genes and eQTLs can be higher level of expression of disease genes, and the non-coding variants would likely have lesser impact on highly expressed genes. Two-fold or higher change in genes expressed at TPM of 0.1 or even 1 may show high statistical significance but has little or no biological relevance. A change of TPM from 100 to 150 (even if statistically less significant) may be more physiologically important.
- (vi) How was the deconvolution of NSR dataset performed? Did the authors use any published single cell datasets? The proportion of cell types is quite aberrant. A normal human retina has over 70% rods, and the number of astrocytes even in older samples is abnormally high.
- (vii) The results and conclusions such as rare variants as drivers of transcriptomic outliers require careful evaluation with high expression genes. This is especially important since no biological validation has been performed.
- (viii) Technical and methodological details are not included for most analyses.

Other comments and suggestions:

1. The authors have mentioned they found 18,891 and 13,214 genes expressed at moderate (TPM >1) and high (TPM > 5) levels in both the tissues. What genes and TPM filter were used for differential expression analysis? How was the data normalized and what covariates were used for correcting for hidden factors and batch effects as the NSR samples RNA sequencing was done in 5 batches and RPE samples in 3 batches?
2. The authors state that expression variability was significantly higher in RPE compared to NSR across genes expressed at low, moderate, and high levels ($p < 2.2 \times 10^{-16}$; Fig. 1B). Could the authors clarify which statistical test was performed to assess this difference? Specifically, was this a group comparison across samples, or a paired analysis between matched RPE and NSR samples from the same individuals?
3. Median age 71 [64,77], Median PMI 40 [32,44], Male predominance [63.7%] presents issues when comparing to Gtex or EyeGex. Male-female differences in gene expression are well documented for various tissues.
4. The authors have not mentioned anything about the ancestry of the samples. The samples are presumably of European

ancestry. Was ancestry check done on the samples before running the QTL analysis?

5. For the WGS sample level quality control the authors found outliers. How many samples were finally used for the QTL analysis for NSR and RPE. The authors also need to provide QC plots to show how they identified the outlier's samples.

6. How did the author decide to use 30 PEERS factors in the QTL analysis. How many total PEERS were identified. Did the authors perform bench marking for the PEER factors to identify the QTLs.

7. The gene ontology results for the differentially expressed genes in Supplementary table 3 and Table 4 has redundant GO terms and genes. The redundant terms can be summarized based on semantic similarity or hierarchical relationships.

8. The authors report 353,385 novel eQTLs in genes previously described by EyeGEx and 429,377 in 3,966 newly identified eGenes. However, it remains unclear what these novel signals are enriched in and whether they reach functional or disease-relevant significance. For AMD risk loci, 26 previously reported eQTLs were examined, of which 15 were replicated or in high LD, with additional novel METR-eQTLs identified for 5 genes. Rather than summarizing in a categorical table format (replicated, in LD, novel, no signal, not replicated), a formal colocalization analysis would have been more informative, as it directly tests whether the same variant explains both expression and disease association. The table-based approach is descriptive but limited in capturing fine-mapping resolution and uncertainty.

9. The authors mention about the replication of eQTLs that impact AMD risk genes and were identified as lead candidates in published eQTLs studies by comparing the eQTLs and checking the variants are in LD, but they have not performed any colocalization analysis with AMD GWAS and eQTL to find this was the target gene identified for each of the loci reported in the Table 1 or to identify novel target genes for AMD GWAS loci with METR-eQTLs.

10. Please clarify the statistical basis for classifying a gene as having 'no METR eQTL. Is this defined solely by the absence of overlapping eQTLs under an FDR threshold of 0.05?

11. Given that EyeGEx had nearly twice the sample size, METR (~200 samples) may be underpowered. At minimum, one could attempt targeted replication of the exact EyeGEx variant-gene pairs using relaxed/nominal thresholds and LD-aware proxies, classifying as 'replicated via LD' if a proxy meets a one-sided nominal p-value in the same direction (after allele harmonization). Where local summary statistics are available (± 1 Mb), formal colocalization (e.g., coloc, eCAVIAR) would provide stronger evidence of shared causal signals, even if the exact lead SNPs differ.

12. What was the r^2 scores in LD for variants identified in the eQTLs reported in Table 1.

13. Lines 110–111: The text states that 13 previously reported AMD-associated eQTLs were replicated (Table 1). However, the legend of Table 1 specifies that 15/26 previously reported eQTLs were replicated or in high LD. This discrepancy should be clarified — is the correct number 13 or 15? Please ensure consistency between the main text and table legend.

14. The authors have compared METR-eVariants to cell-type agnostic cis-candidate regulatory elements (cCRE) available through ENCODE and found METR-eVariants were enriched in promoters and proximal enhancers compared to control variants matched for allele frequency and gene density. The variants which are reported in Table 1 for AMD GWAS they were enriched in which regions.

15. Figure 4 suggests that gene regulation most relevant to disease may not be driven primarily by bulk expression levels, but instead by other regulatory mechanisms such as splicing. Indeed, eGenes linked to EyeG2P disease genes show lower coefficients of variation and smaller absolute effect sizes. While the authors interpret these findings as consistent with the general observation that common eVariants tend to have smaller effects than rare ones, their additional claim of selective bias against rare eVariants in eye disease genes is overstated without direct population-genetic evidence (e.g., constraint metrics, allele frequency depletion analyses, evidence that depletion is specific to eyeG2P genes vs matched random or permuted control set)

16. The rare variant analysis includes splicing and allelic imbalance outliers, yet the study only assessed expression QTLs. Why were sQTLs and ASE not evaluated? If the goal is to assess the power and contribution of rare variant analyses to transcriptomic regulation, then eQTLs, sQTLs, and ASE should all be systematically compared. It would be valuable to quantify how common versus rare variants mediate transcriptomic variation (expression, splicing, allelic imbalance) and to report the proportion of variance explained in each category, as this would provide a clearer picture of their relative impact on disease mechanisms.

17. Please define the method or criteria used to classify eOutliers as 'co-occurring' with SVs/CNVs. For example, is this defined as an outlier expression event for a gene in an individual overlapping with a rare variant in the same gene in the same individual? Please clarify the method, definition, and thresholds.

18. Lines 175–176: Please define what constitutes an 'outlier event.' Is this defined as a gene-individual pair with expression significantly deviating from the expected distribution? Additionally, clarify the cutoff used — was this based on Z-score thresholds, residual modeling, or strictly on an adjusted p-value < 0.05 from the aberrant expression module?

19. Consider defining an explicit Z-score threshold for calling outlier events, rather than relying only on adjusted p-values. For example, using both $p_{adj} < 0.05$ and $|Z| > 2$ could help benchmark sensitivity (e.g., at ~1:5 outlier-to-gene ratio) and

make the definition of outlier events more interpretable. Additionally, it would be valuable to present enrichment analyses by minor allele frequency bins, stratified across variant classes (SNVs, indels, CNVs, SVs), to better contextualize how rare versus common variants contribute to outlier expression.

20. Table 2: The phrasing ‘median number of impacted genes per sample across expression, splicing, and allelic balance’ is not very intuitive in meaning. Did the authors first check whether specific samples were outliers that might be disproportionately driving these events?

21. Line 194: The phrasing ‘METR outliers’ should be revised to ‘outlier events.’ Referring to them simply as ‘outliers’ risks implying that an entire sample is an outlier, whereas the analysis is performed at the level of specific gene–sample outlier events.

22. Lines 197–199: Please specify the method used to link unexplained eOutlier events with cCREs. For example, was the approach: for each unexplained outlier event, determine whether the same individual carries a rare variant (SNV, indel, SV, CNV) within 10 kb of the gene body that overlaps an eye-specific cCRE (Cherry et al., 2020)?

23. Lines 782–797: The definition of shared eQTLs relies on exact matches of variant ID (CHR_POS_REF_ALT, GRCh38) and Ensembl gene ID (without version). Please clarify how harmonization was performed across datasets to ensure consistency in variant calling and gene annotation. For EyeGEx replication, you extracted top eQTLs and checked for replication or high LD ($r^2 > 0.8$) using LDlinkR — it would help to specify whether allele harmonization (strand/orientation) was also conducted.

24. The manuscript should explain how non-genetic drivers of widespread extreme expression were handled — for example, by removing individuals with outliers in >50 genes from downstream analysis! The authors note that excluding such individuals improved rare variant enrichments, but it’s unclear how this threshold was chosen and whether the results are robust to alternative cutoffs. Additionally, do the authors have replication for the identified eOutliers? The methods section states that enrichment was assessed only in genes with at least one outlier and one control, and that relative risk was estimated for the presence of a nearby rare variant given outlier status (with 95% Wald CIs). Explicitly describing the replication strategy, alongside this enrichment framework, would make the conclusions more convincing.

25. Please define explicitly what constitutes an ‘outlier’ in this ASE analysis. Does this include both underexpression and overexpression events, or only one direction? The text suggests expectations for both (minor allele underexpressed in underexpression outliers, overexpressed in overexpression outliers), but the definition of outlier status and how it was assigned (e.g., based on Z-scores, adjusted p-values, thresholds) should be clarified.

26. The DROP workflow is currently used for rare disease diagnosis purposes so how is this method useful in complex diseases such as AMD? It is not clear what are the author’s trying to convey with the list provided in Supplementary table 5.

27. Authors have not explained what expression outliers are in Result section.

28. The methods Figure 1, Supplementary figure 3, Figure 6 and 7: the font size is too small and hard to read. The figures also get blurred when tried to enlarge.

Version 1:

Reviewer comments:

Reviewer #2

(Remarks to the Author)

The manuscript has been revised. However, the major concerns have not yet been thoroughly addressed. For example, a meta-analysis integrating the data with other available datasets has not been performed, and the manuscript does not clearly demonstrate how this study provides novel insights beyond existing work. Although authors state that the study has sufficient statistical power to detect regulatory effects across approximately 10,000 genes, similar or more comprehensive analyses have already been reported in at least three independent studies.

Second, the study identifies only a very small number of rare-variant eQTLs. While authors note that this number is comparable to previous report, the findings remain modest. Consequently, the utility of this dataset for robust investigation of rare variant–driven regulatory mechanisms is limited.

Reviewer #3

(Remarks to the Author)

Thanks for responding to my comments and suggestions. I broadly agree with the response (though we may have a couple of scientific disagreements) and commend the authors for a thorough revision.

Anand S.

Reviewer #1 (Remarks to the Author):

The manuscript by Sampson, et al. utilizes whole genome sequencing and RNA-Seq across ~201 human donor eyes to better understand gene expression regulation in the neural retina and RPE. They identify nearly 1.5 million eQTLs that affect over 13,000 genes in the retina and RPE. Further, they identify expression outliers that can be attributed to SNPs, structural and copy number variants. The authors attempt to make this data translatable to ocular disease, in this case, specifically to AMD. Overall, the research strategy is sound, however, there are major and minor concerns that need to be addressed.

Major concerns

1. Average PMI of 40 hours likely introduces a lot of artifacts in the RNA-Seq data. This needs to be addressed in the Discussion.

We thank the reviewer for raising this important point and have included additional data in the main results, and additional supplementary figures confirming minimal impact of PMI time on RNA integrity and 3'/5' bias (**Supplementary Figures 1 and 4**). We have also included **Supplementary Table 3** with a range of QC metrics. In summary, RNA Integrity Number (RIN) was available for 181/183 NSR samples and 169/176 RPE samples, with a median RIN for NSR samples of 7.9 (IQR=7.5,8.1), and a median RIN for RPE of 6.9 (IQR=6.5,7.5). There isn't a noticeable decrease in RIN scores in samples with high PMI time (> 40 hours; **Supplementary Figure 1**). We have added description of the approaches to the methods, data to the main results section and supplementary figures and tables, and added text to the 2nd paragraph of the discussion to highlight this important point and give additional context.

2. Given the high PMI, the average RIN should be given for the samples. They are likely below 7, which is the baseline for RNA-Seq. This does not mean the samples are invalid, but high 5' bias will likely affect the results

Please see our response to the point #1, which addresses some of these points raised by the reviewer. In addition, we have assessed 3'/5' biases in the RNAseq data generated, and thank the reviewer for the opportunity to include this in the manuscript. In summary, the median 3'/5' bias in NSR samples was 0.5 (IQR=0.48-0.51), and in RPE was 0.51 (IQR=0.49-0.55). We have added the approaches for this calculation to the methods, data to the main results section and supplementary figures (**Supplementary figure 1B**), and added text to the discussion in response to this point.

3. Similarly, using 25 ng to 1 µg of RNA for library preparation is not a good practice. This is because the samples will require significantly different number of PCR cycles. The lower yield samples will have more duplicates and less representation of lower expressing genes.

We thank the reviewer for raising this point and enabling increased clarity in the approaches undertaken. Whilst the protocol for the Illumina Stranded mRNA ligation

preparation allows samples between 25ng-1µg, we restrict this range to 200-500ng and performed a fixed number of 12 PCR cycles during the preparation. All samples were then normalised prior to loading on the sequencers. We have added these details to the methods section of the manuscript to provide clarity of the approaches undertaken.

Moreover, we have assessed our datasets to ensure the integrity of the datasets created, observing an average of 147 million (IQR=138-161 million) reads for NSR with an average unique mapping rate of 89.4% (IQR=88.2-90.7%) and an average of 74 million (IQR=59-94 million) reads for RPE, with an average unique mapping read of 82.9% (IQR=78-86%). These data show that our samples were sequenced at high depth and have a relatively low proportion of duplicate reads.

We have added approaches to the methods, data to the results and supplementary figures (**Supplementary figure 4**) and tables (**Supplementary table 3**), and added text to the discussion to address this point.

4. Based on the Methods, it is not clear that duplicate reads were removed from the data prior to quantification.

We thank the reviewer for the opportunity to clarify the methodologies undertaken in this study. In summary, we applied the GTEX v8 RNASeq pipeline for processing of RNASeq data (GTEx Consortium, 2020). In this pipeline, duplicate reads are annotated following read alignment using Picard v.2.27.1. The annotated BAM files are then processed with RSEM (Li and Dewey, 2011) and RNASeQC (DeLuca *et al.*, 2012) for gene expression quantification. The GTEX authors recommend this approach due to ambiguities in resolving the source of duplicates, which can be biological or technical (<https://github.com/broadinstitute/gtex-pipeline/issues/29>). We have added clarity to the methods about this approach. Please also see response to point #3.

5. The differential expression algorithm used should be listed in the methods.

We thank the reviewer for raising this point and have added a detailed description of the differential expression analysis in the methods section. *[Please also refer to Reviewer 3, comment #4]*. In summary, we applied approaches utilising the *deseq* R package (Love, Huber and Anders, 2014) and corrected for age and sex as covariates, and *edgeR* (Robinson, McCarthy and Smyth, 2010).

6. It's mentioned that 49 eyes have AMD risk SNPs, but no pathology. Have other common diseases been ruled out (glaucoma, diabetic retinopathy, etc.)?

The Manchester Eye Tissue Repository is an anonymous tissue bank resource with no direct link to detailed clinical records for donors - this is required within the terms of the ethical agreement for the repository. Whilst limited clinical information is available for donors when provided on the organ donation forms, this information cannot be ratified, is not comprehensive and does not exclude the presence of

disorders which may impact eye health. From this limited clinical information provided on the donation forms, we actively selected against donors that were likely to have other retinal issues. To further ensure that the cohort comprised individuals with 'healthy' retinas, detailed post-mortem phenotyping was performed for all donors to assess for the presence of drusen and other signs of AMD pathology and retinal health (Mcharg *et al.*, 2022). These workups confirmed absence of late-stage AMD, inherited retinal disorders and other retinal related disorders such as advanced diabetic retinopathy.

7. Since such a large focus was placed on translation to AMD, it seems that neither the macula RPE, or retina were sampled. It's been shown that the macula RPE differs transcriptionally from the peripheral RPE. And no cones were among the list of deconvoluted cell type of the retina. This makes the AMD association difficult.

We thank the reviewer for raising this important point, and have elaborated our discussion of this in the manuscript. We agree that the transcriptional patterns of macula retina and RPE are different to the peripheral tissue, as has been shown in Li *et al.* (2014). We also agree that alternative methods could have been adopted in this study, e.g. foveal staining with targeted biopsy or single-cell approaches. The lack of fresh frozen tissue, the availability of samples (the RPE is only available from the eye tissue bank as a single pelleted sample containing both peripheral and central RPE), and the opportunity to develop whole genome sequencing data and RNAseq from both NSR and RPE were key considerations in our study design. The bulk and high-depth RNA sequencing approach adopted in this study includes both macular and peripheral tissue and we expect to have captured eQTL associations reflecting both parts of the eye, and whilst we agree with the reviewer that other complementary approaches would enable unpicking of eQTLs that differ between the central and peripheral regions of the retina, this was not a primary objective of this study.

Minor comments

1. It's unclear why IRDs are mentioned in the Introduction. This leads the reader to think they'll be discussed with the data but are not.

We agree with the reviewer that the link between IRDs and the analysis described to investigate the properties of eVariants and eGenes was not clear enough (**Figure 4 & Supplementary figure 11**). We have rectified this and added description in the methods, and added clarity to the results and discussion.

2. The authors mention that higher depth of sequencing was achieved in the current study compared to those in the literature. In all studies, the numbers generated for the RPE likely reach 90% or greater depth of coverage. If the authors would like to keep that statement in the Discussion, then they should provide depth of coverage plots for their samples.

We thank the reviewer for the opportunity to clarify this point – the statement provided in the discussion was intended to refer to the NSR samples rather than RPE, although we do also generate increased coverage for the RPE compared to previously described studies. Compared to previous studies (Ratnapriya *et al.*, 2019;

Orozco *et al.*, 2020; Strunz *et al.*, 2020) our data resource provides 4.5-fold and 2.4-fold increase in RNA sequencing data for NSR and RPE, respectively. A selection of these are included below:

- The median sequencing depth of NSR in EyeGex was 32.5 million,
- Orozco *et al.* sequenced 30 million single end 50bp reads in NSR and RPE
- Strunz *et al.* (2020) sequenced in different batches, with read depth for NSR varying between 20 and 80 million reads).

To ensure this is clear we have refined the section of the discussion discussing this point.

3. The reporting summary states that the data will be available through dbGAP, yet this is not mentioned in the manuscript.

We apologise for the delay in making this data available. All raw data has been shared in the European-Genome Phenome Archive (Study ID: EGAS50000001443; Dataset: EGAD50000002082). This includes FASTQ files from whole genome sequencing, BAM files from RNA sequencing for both NSR and RPE, as well as processed variant calls, gene quantification statistics, eQTL results, eOutlier statistics and aggregated datasets. All paired samples are linked by a unique study ID for each participant.

Reviewer #2 (Remarks to the Author):

Genetic variation influencing retinal gene expression is still not fully understood, particularly in the context of vision-related disorders. In this study, the authors performed whole-genome and RNA sequencing on neurosensory retina (NSR) and retinal pigment epithelium (RPE) samples from 201 post-mortem eyes. They identified over 1.4 million significant cis-eQTLs as well as 299 rare variants that may contribute to expression outliers. These findings advance our knowledge of both common and rare genetic regulation in human retinal tissues. This reviewer has the following comments for the authors to consider:

1. Numerous studies, including GTEx, have consistently demonstrated that the discovery power for eQTLs increases substantially with larger sample sizes. The authors correctly note that a large-scale eQTL study in the retina has already been published. However, in the present study, they choose to analyze their dataset of 200 retina samples independently and only compare their results with the previously published dataset. This approach represents a missed opportunity. By combining their data with the existing dataset, the authors could have achieved a significantly larger and more statistically powerful sample size, potentially uncovering additional and more robust eQTLs. I strongly recommend that the authors consider performing a joint or meta-analysis with the previously published data. Such an integrative approach would enhance the impact and biological insight of the study.

We thank the authors for noting the uniqueness of this resource and the rarity with which robust scientific investigation has enabled the discovery of genetic variation impacting gene expression in the retina. We agree that a meta-analysis combining datasets from previously published studies is of high interest for future work, including data published by Ratnapriya et al (2019) and Strunz et al (2020). Combining such datasets would increase sample sizes to detect eQTLs, and is predicted to increase estimated power of previously reported datasets. However we note that our study design was adequately powered to detect over 10,000 eGenes across NSR and RPE, and capable of detecting eVariants with internal variant population frequencies as low as 2.5% and effect sizes ranging between <0.2 and $6.6 \log_2$ absolute allelic fold change in NSR and RPE (**Supplementary figure 7**).

The approaches undertaken by previous studies differ in sample preparation, RNA sequencing depth and read length, genotyping approach, and bioinformatics methods applied. As a result, at a minimum, meta-analyses will require extensive harmonisation of the bioinformatics methods applied to both RNA sequencing and DNA genotyping approaches and control for confounding issues specific to the sample preparation and RNA sequencing approaches undertaken. The datasets that are required for this significant undertaking are not publicly available to the best of our understanding, and as such was not included in our study design. Further, whilst we expect that combining datasets may enable increased power to detect eQTLs, it would not increase sample sizes to detect rare variants and structural variants that impact gene expression in the retina, which is a key and unique advantage of the whole genome sequencing approaches taken in this study.

In summary, we have added additional data to the manuscript to confirm the range of eQTLs that were identified (**Supplementary figure 7**), and added increased context

to the discussion. We look forward to collaborating with other teams to overcome the significant challenges associated with a meta-analysis of previously published retina and RPE studies, but confirm that it is out-of-scope for the analyses reported in the current study, and are confident that the biological insights and findings from the current study are justified as a stand-alone publication.

2. The comparison of METR-NSR eQTLs for each eGene identified in this study with the retina-specific eQTL dataset from the EyeGEx project is not entirely appropriate. The EyeGEx study focused exclusively on gene expression in the neural retina, whereas the METR-NSR dataset includes eQTLs derived from both the neural retina (NSR) and the retinal pigment epithelium (RPE), either separately or in combination. As a result, the eQTLs identified in METR-NSR may reflect regulatory variation specific to the RPE, which would not be captured in a retina-only dataset like EyeGEx. To ensure a more accurate and biologically meaningful comparison, the authors should separate the RPE-derived eQTLs from those identified in the NSR and restrict their comparison with EyeGEx to the NSR component. This would allow for a more valid one-to-one comparison and avoid confounding results due to tissue-specific regulatory differences.

We agree with the reviewer that the comparison of our eQTL dataset with EyeGEx should exclusively focus on the overlap between eQTLs identified in the neural retina. The comparison with EyeGEx initially reported in the manuscript exclusively focussed on eQTLs identified from the neurosensory retina (METR-NSR), and we have reworded this part of the results and amended the legend for **Figure 2B** to ensure clarity. In the revised manuscript we have also extended the comparison with eQTLs identified by Strunz et al. (2020) which also focus on expression in the neural retina, and have therefore limited comparison to the METR-NSR eQTLs. These analyses have been added to the results, including **Figure 2B** and to **Supplementary figure 8** and to the discussion. Please also see response to reviewer #3, comment #1.

3. In this study, the authors have analyzed cell-type agnostic cis-candidate regulatory elements (cCREs) as defined by ENCODE (v3). While ENCODE provides a valuable and comprehensive resource, it is based on broadly defined regulatory elements that may not capture the specificity of gene regulation in ocular tissues. Importantly, several high-quality studies have been published that provide retina- and RPE-specific chromatin accessibility and regulatory element datasets, including those by Wang et al. (2022), Orozco et al. (2020), Wang et al. (2023), and Cherry et al. (2020). These resources offer cell-type or tissue-specific regulatory insights that are more biologically relevant to the context of this study. I strongly recommend that the authors incorporate these eye-specific datasets into their analysis to derive more nuanced and biologically meaningful interpretations. Doing so could enhance the resolution and relevance of their regulatory annotations, particularly in understanding tissue-specific gene regulation mechanisms in the retina and RPE.

We thank the reviewer for highlighting the importance of integrating relevant tissue-specific regulatory annotations with our eQTL data. In the initial manuscript, we

incorporated retina and RPE-specific candidate cis-regulatory element (cCRE) annotations from Cherry et al. (2020) and used bootstrap enrichment analysis to calculate the relative enrichment of eQTL variants which overlapped with each type of regulatory element (**Figure 3**). Following this approach, we show a significant enrichment of METR-eQTL variants in retina-specific (p-value = 4.52×10^{-28}) and RPE-specific cCREs (p-value = 8.74×10^{-10}) compared to the control variants. Non-eye cCREs from adult tissues in EpiMap were also enriched for METR-eVariants, although the enrichment was lower than in the NSR and RPE (**Figure 3B**).

We agree that extending these additional available datasets may provide extra insights and in the revised manuscript have compared eQTLs to cell-specific accessible chromatin regions detected by single cell ATACseq from retina tissues (Wang et al., 2022 and Wang et al., 2023). We applied the same bootstrap enrichment analysis approach that we performed for Cherry et al. (2020) datasets and identified enrichment of METR-eQTL variants in scATACseq peaks in all cell types queried: rod cells (p-value = 6.69×10^{-58}), cone cells (p-value = 6.58×10^{-57}), astrocytes (p-value = 7.78×10^{-25}), horizontal cells (p-value = 2.31×10^{-21}) retinal ganglion cells (p-value = 2.21×10^{-20}), Muller glia (p-value = 5.35×10^{-19}), bipolar cells (p-value = 7.02×10^{-18}) and amacrine cells (p-value = 2.08×10^{-6}). The analyses have been added to the methods, results, main figures (**Figure 3C**) and discussion of the revised manuscript.

4. While the use of whole-genome sequencing (WGS) data to investigate the role of rare variants in gene expression regulation is innovative and represents a valuable direction for the field, the current study appears significantly underpowered for such an analysis. The sample size is relatively small for detecting rare variant effects, which typically require much larger cohorts to achieve sufficient statistical power. As reported, only 1,051 expression outliers were identified, and among these, just 299 rare non-coding single-nucleotide, structural variants or copy number variants as plausible for 28% of outlier events were prioritized as plausible contributors—accounting for approximately 28% of the outlier events. While this is a noteworthy effort, the modest number of supported associations limits the strength of the conclusions that can be drawn. As a result, the overall enthusiasm for the utility of this dataset to robustly explore rare variant-driven regulatory mechanisms remains tempered.

We thank the reviewers for raising this point and agree that larger sample sizes are more powered to detect rare variants driving outlier expression, but we do not consider our dataset to be underpowered for this analysis. We identified 1,171 expression outliers (eOutliers) in NSR and RPE, of which 1,028 were unique (728 in NSR, 443 and 120 in both tissues). These eOutliers were identified using the Aberrant Expression module of the DROP pipeline. The developers of this tool estimate that the minimum number of samples per tissue required to detect expression outliers is 50 (<https://gagneurlab-drop.readthedocs.io/en/latest/prepare.html>). Moreover, we compared our findings with a recent study by Hölzlwimmer et al. (2025) that investigated aberrant expression in the GTEx cohort. The authors determined that 0.026% of gene-sample events were significant outliers, compared to 0.023% and 0.015% achieved for NSR and RPE in this study, respectively. We agree that this additional context helps

clarify the appropriateness of the data generated in this study and have added this to the discussion section of the manuscript.

5. The authors state that the METR resource can be integrated with other multi-omic datasets to support the discovery of novel eye-specific regulatory elements, including those potentially involved in both common (e.g., AMD) and rare (e.g., IRDs) genetic disorders affecting the retina. While this is an important and promising application of the resource, the current manuscript does not present any direct evidence or in-depth analysis specifically demonstrating the utility and power of METR for studying rare variant contributions to IRDs. Given the complexity and low allele frequencies typically associated with rare disease-causing variants, more focused analyses or case studies—linking METR data with known IRD loci or variant annotations—would help substantiate this claim. As it stands, the utility of METR for rare variant interpretation remains speculative. Figure 6 and 7 present selected examples, but doesn't not provide much functional relevance. Given the challenges associated with interpreting the regulatory impact of rare, non-coding variants, functional follow-up is essential to establish biological relevance. Including such validation—whether through reporter assays, CRISPR perturbations, or integration with existing disease-linked variant databases—would greatly strengthen the authors' claims and enhance the translational potential of the METR resource for rare disease research.

We agree with the reviewer that functionally validating the impact of rare variants on gene expression through orthogonal techniques enhances the translational potential of our resource for both rare disease and AMD research. In the original **Figure 6C**, we provided an example of a rare variant detected in the promoter of *CAND2* in two donors with a strong decrease in gene expression in the neurosensory retina and RPE (*NM_001162499.2:c.-41A>G*). In response to the reviewers suggestions, we have designed a dual reporter luciferase assay to investigate the impact of this promoter variant on downstream expression. The relative luminescence detected in cells carrying the rare promoter variant (*c.-41A>G*) was 50% lower than in cells carrying the wild-type allele and >50% lower than in cells carrying a nearby and expected benign variant (*c.-36C>T*) that is common in the gnomAD resource. We have added these investigations to the methods, results, main figures (**Figure 7**) and discussion.

We also agree that clearer clarification of the work conducted in this manuscript relevant for rare disease research would be of benefit. In line with the response to reviewer #1 minor comment #1, we have revised and amended results and figure legends to ensure that the investigations related to rare monogenic eye disorders are more clearly articulated. We envisage that this data will be used by our team, and others, in future work to assist in the identification of rare non-coding variants disrupting gene expression. This work is outside the scope of this manuscript but relevant investigations have been published elsewhere by our team, including recommendations for variant interpretation, discovery of novel non-coding disease genes and identification of novel genomic variants in untranslated regions, enhancers and promoters of known disease genes. These analyses, and others, are referred to in the discussion section to set the context for additional work that could be undertaken.

6. The authors claim that the higher sequencing depth used in their study leads to improved gene detection and expression quantification. Specifically, they report that compared to EyeGEx, their high-coverage approach increases the number of detectable protein-coding genes by 23% (from 13,662 to 16,765) and newly identifies 3,663 eGenes. It would be important to clarify whether these gene counts and newly identified eGenes are derived exclusively from neural retina (NSR) samples, or whether they also include data from the retinal pigment epithelium (RPE), which could confound direct comparisons with EyeGEx, a retina-only dataset. Additionally, there appears to be a discrepancy in the reported overlap with EyeGEx findings. EyeGEx identified approximately 10,000 eGenes, but, yet the current study reports overlap with only 5,993 NSR eGenes from EyeGEx. The authors should address this gap and provide an explanation as to why a substantial number of previously reported eGenes were not detected in their dataset. Potential contributing factors—such as differences in sample size, statistical power, expression thresholds, or tissue composition—should be discussed to help contextualize this observation

We thank the reviewer for the opportunity to clarify the comparison between our eQTL dataset and EyeGEx. We confirm that only the eQTLs identified from the neurosensory retina (NSR) were included in the comparison. Specifically, of the 9,959 genes associated with an eQTL variant (eGenes) in the METR-NSR, 5,993 were also identified by EyeGEx. Consequently, 3,966 eGenes were only identified in our study, and 4,481 were only identified by EyeGEx. In response to this comment we have added data to the results (**Supplementary figure 8**) and added a section to clarify this context in the discussion.

Minor points

1. Page 12, lines 288–293, contains a repetitive statement about sequencing depth that was already addressed in the second paragraph of the Discussion. The authors may consider removing or consolidating this section to avoid redundancy.

We have removed the lower-depth statement in the second paragraph of the discussion.

2. The manuscript currently lacks a statement regarding data availability. Could the authors please include information on how and in what form the data will be shared, including any relevant accession numbers or repositories?

We have included a statement about data availability. Please also refer to the response to Reviewer #1 minor comment #3 for full details of the data available.

Reviewer #3 (Remarks to the Author):

This manuscript describes analyses of whole genome sequencing and bulk RNA-sequencing of neurosensory retina (NSR) and retinal pigment epithelium (RPE) from 201 post-mortem eyes. The authors have reported differential expression analysis of genes (NSR vs RPE), identified eQTLs for NSR and RPE, and compared these to published retina eQTLs, GTEx eQTLs and eGenes. The authors have also integrated cis-regulatory elements (cCREs) from ENCODE and detected expression outliers and prioritized 299 rare non-coding single-nucleotide, structural variants or copy number variants.

Overall, the authors have attempted to put together a good resource for retina and RPE transcriptome and eQTLs. However, there are numerous issues pertaining to analyses and significance. If the analyses are performed using similar parameters, it is unclear how this study would compare with what has already been published with larger study samples (see comments below).

Major issues:

(i) Small number of retina samples (about 200) compared to Ratnapriya et al. Nat Genet 2019 which included 453 controls and AMD cases. Another paper Strunz et al. PLoS Genet 2020 that was not included by the authors used 311 healthy human retina samples for eQTL analysis. Thus, the study is underpowered for eQTL analyses compared to published studies, at least for the retina.

We acknowledge that the sample size in this study is smaller than both Ratnapriya et al. (2019) and Strunz et al. (2020), but we note that the study was adequately powered to detect over 10,000 eGenes across NSR and RPE, and capable of detecting eVariants with internal variant population frequencies as low as 2.5% and effect sizes ranging between <0.2 and 6.6 log₂ absolute allelic fold change in NSR and RPE (**Supplementary figure 7**).

The approaches undertaken by previous studies also differ substantially in RNA sequencing and genotyping techniques. Whilst a number of other approaches could have been adopted in this study, the opportunity to expand analyses to rare variants through whole genome sequencing was a primary motivation and a key factor in the study design. As such, we develop a first-of-its-kind resource and expand understanding of the types of genetic variants and regions that regulate/disrupt gene expression in NSR and RPE.

In response to this comment we have added data to the results (**Supplementary figures 7 & 8**) to demonstrate the range of eQTLs that were detected and how they compare against the findings described by both Ratnapriya et al. (2019) and Strunz et al. (2020).

(ii) RPE samples have greater variability (line 67) because of biological and technical reasons. RNA from older RPE samples is generally degraded. The authors have not provided any information with respect of sample collection, RNA quality and other

variables, which can significantly impact the expression variability. It's unclear if this data is trustworthy.

We thank the reviewer for raising this important point and in response, we have included an additional and extensive supplementary table with relevant QC metrics (e.g. RNA Integrity number, the number of mapped reads, read length, 3'/5' bias and PMI time), as well as additional figures associated with the sample resource, including the impact of PMI time on key metrics (**Supplementary figures 1 & 4**). RNA Integrity Number (RIN) was available for 181/183 NSR samples and 169/176 RPE samples, with a median RIN for NSR samples of 7.9 (IQR=7.5,8.1), and a median RIN for RPE of 6.9 (IQR=6.5,7.5). There isn't a noticeable decrease in RIN scores in samples with high PMI time (> 40 hours; **Supplementary Figure 1**). In summary, these data show there is minimal impact of PMI time on the quality of the data produced.

We have added description of the approaches to the methods, data to the main results section and supplementary figures and tables, and added text to the 2nd paragraph of the discussion to highlight this important point and give additional context.

(iii) Both Ratnapriya and Strunz studies used more conservative expression values of CPM>1 or >2 in 10 or 50% of samples. This study appears to have selected TPM values of >0.1 for subsequent analyses resulting in 28,512 genes. This is a rather high number. It's unclear if a change in TPM of 0.1 to 0.2 or 0.5 (which are 2-5-fold change) even if statistically significant would have any biological relevance. At the very least, the authors need to repeat the complete eQTL analysis with TPM of >1 in 10-20% of samples and with lesser number of genes.

We would like to thank the reviewer for the opportunity to clarify the input expression thresholds used for eQTL mapping. We have included a detailed description of the thresholds we used in the methods. Briefly, for each tissue, genes which did not meet expression thresholds of >0.1 TPM in at least 20% of samples and ≥ 6 reads in at least 20% of samples were removed from eQTL analysis. This was in accordance with the methods established by the GTEx consortium (<https://www.gtexportal.org/home/methods>), considered as gold-standard for eQTL mapping from bulk-RNAseq data. This approach resulted in 26,734 genes in neurosensory retina and 24,448 genes in retinal pigment epithelium for analysis, and provided a standardised and widely established method for direct comparison against eQTLs characterised for other tissues (**Figure 2C & Supplementary figures 9 & 10**).

(iv) The authors have identified 21,157 differentially expressed genes with adjusted p value of 0.01 between NSR and RPE. The number is too high and unreasonable. The significance of this differential expression analysis between neural retina and RPE is unclear. These are two completely different tissues, and such large differences in gene expression do not add much value. Common genes will likely be constitutively expressed genes. Even otherwise, how was the differential expression analysis done (the methods are missing)?

We thank the reviewer for raising this point and the opportunity to clarify the context of this analysis. We have added a detailed description of the differential expression analysis in the methods section. In summary, we applied approaches utilising the *deseq* R package (Love, Huber and Anders, 2014) and corrected for age and sex as covariates, and *edgeR* (Robinson, McCarthy and Smyth, 2010).

We would like to emphasise that the main purpose of the differential expression analysis in our manuscript is to ensure that the transcriptomic data for NSR and RPE is reflective of biological pathways we expect to be active in each tissue. We utilise this method, and describe the findings in the manuscript to confirm the integrity of the data resource and have added clarification of this intention to the methods, results and discussion.

We acknowledge that the number of differentially expressed genes we reported may seem unusually large, and is likely impacted by different tissue context and the different levels of coverage achieved for NSR and RPE, and agree that including reference to this in discussion may cause misinterpretation of it's intention. We have removed and refined the discussion of this point to ensure we are explicit about the intentions and limited conclusions that can be drawn from these analyses.

(v) The simple reason for disease and non-disease genes and eQTLs can be higher level of expression of disease genes, and the non-coding variants would likely have lesser impact on highly expressed genes. Two-fold or higher change in genes expressed at TPM of 0.1 or even 1 may show high statistical significance but has little or no biological relevance. A change of TPM from 100 to 150 (even if statistically less significant) may be more physiologically important.

We agree with this point and agree that genes associated with rare monogenic eye-disease (disease genes) have higher expression in retina than non-disease genes. In response, we have refined this analysis to control for expression level of known disease genes and described in the methods, results (**Supplementary figure 11**) and discussion. In summary, we still observe differences in key properties of eQTLs associated with disease-genes compared to non-disease genes when controlling for mean gene expression.

(vi) How was the deconvolution of NSR dataset performed? Did the authors use any published single cell datasets? The proportion of cell types is quite aberrant. A normal human retina has over 70% rods, and the number of astrocytes even in older samples is abnormally high.

We thank the reviewer for the opportunity to clarify the methodology used to perform the deconvolution analysis. A detailed description is included in the methods section. Briefly, we used the *BayesPrism* R package (Chu *et al.*, 2022) to run a deconvolution model to estimate the proportion of retinal cell types from our bulk-RNAseq data in the NSR and RPE. To do this, we used publicly available single-cell RNAseq data from the ocular posterior segment published by Monavarfeshani *et al.* (2023)

(Available for download from https://singlecell.broadinstitute.org/single_cell/, study number: SCP2310).

We agree that the high level of astrocytes is an unusual observation and have extended our comments and context in the discussion to explain some reasonings for this.

(vii) The results and conclusions such as rare variants as drivers of transcriptomic outliers require careful evaluation with high expression genes. This is especially important since no biological validation has been performed.

We agree with the reviewer that outlier analyses do require careful evaluation and consideration. In the original **Figure 6C**, we provided an example of a rare variant detected in the promoter of *CAND2* in two donors with a strong decrease in gene expression in the neurosensory retina and RPE (*NM_001162499.2:c.-41A>G*). In response to the reviewers' suggestions, we have designed a dual reporter luciferase assay to investigate the impact of this promoter variant on downstream expression. The relative luminescence detected in cells carrying the rare promoter variant (*c.-41A>G*) was 50% lower than in cells carrying the wild-type allele and >50% lower than in cells carrying a nearby and expected benign variant (*c.-36C>T*) that is common in the gnomAD resource. We have added these investigations to the methods, results, main figures (**Figure 7**) and discussion.

(viii) Technical and methodological details are not included for most analyses.

We apologise for the missing methodological details, and have extended these to represent the analyses that are reported in the manuscript.

Other comments and suggestions:

The reviewer has provided a number of very useful and forward-thinking suggestions, and we thank them greatly for these. Broadly, they can be split into: (1) those that suggest high interest for additional analysis to be performed from the datasets; (2) those that require clarification / justification of the approaches undertaken in the current manuscript, which also largely relate to major comment #viii, and (3) requests for increased information about eOutlier event analysis. We have split the comments into these groups to enable more streamlined responses.

(1) Requests for additional analysis

We appreciate the high interest that the reviewer has shown in our work, and the enthusiasm and suitability of this resource to generate increased understanding of the control of gene expression in the human retina and RPE, through extra analysis. As noted by other reviewers and in our initial submission, these datasets are available through EGA. Whilst several of these questions are of interest to my

research team, the sharing of our data will help enable other groups to perform their own analysis, which are out of scope for this first description of the resource.

8. The authors report 353,385 novel eQTLs in genes previously described by EyeGEx and 429,377 in 3,966 newly identified eGenes. However, it remains unclear what these novel signals are enriched in and whether they reach functional or disease-relevant significance. For AMD risk loci, 26 previously reported eQTLs were examined, of which 15 were replicated or in high LD, with additional novel METR-eQTLs identified for 5 genes. Rather than summarizing in a categorical table format (replicated, in LD, novel, no signal, not replicated), a formal colocalization analysis would have been more informative, as it directly tests whether the same variant explains both expression and disease association. The table-based approach is descriptive but limited in capturing fine-mapping resolution and uncertainty.

9. The authors mention about the replication of eQTLs that impact AMD risk genes and were identified as lead candidates in published eQTLs studies by comparing the eQTLs and checking the variants are in LD, but they have not performed any colocalization analysis with AMD GWAS and eQTL to find this was the target gene identified for each of the loci reported in the Table 1 or to identify novel target genes for AMD GWAS loci with METR-eQTLs.

Whilst we agree that co-localisation analysis is one of several approaches that could be adopted to provide additional evidence for the gene-variant pairs identified in our analyses, there are several other alternatives, e.g. transcriptome-wide association studies, fine-mapping and functional wet lab approaches. We adopted to take a wider eQTL detection approach and to assess independent replication (considering linkage disequilibrium) in other published studies as additional evidence for the eQTL (GTEx, EyeGEx, and Strunz et al). Detailed follow-up of individual loci are out of scope of this current manuscript and to ensure clarity to a broad readership and appropriate context for the analyses, we have added these important points to the discussion section of the manuscript.

11. Given that EyeGEx had nearly twice the sample size, METR (~200 samples) may be underpowered. At minimum, one could attempt targeted replication of the exact EyeGEx variant-gene pairs using relaxed/nominal thresholds and LD-aware proxies, classifying as 'replicated via LD' if a proxy meets a one-sided nominal p-value in the same direction (after allele harmonization). Where local summary statistics are available (± 1 Mb), formal colocalization (e.g., coloc, eCAVIAR) would provide stronger evidence of shared causal signals, even if the exact lead SNPs differ.

We agree that other statistical approaches could have been adopted to assess replication of eQTLs, including those that utilise a nominal statistical threshold. The approach that we applied in the manuscript intended to identify independent replication of eQTLs that reach genome-wide significance in the independent datasets, and is thereby enriched for those candidates that are most likely to be true eQTLs.

Further, to ensure clarity in novel signals identified in our datasets, we have assessed eGenes that are uniquely identified in each resource (EyeGEx and Strunz

et al) at genome-wide significance. We have added text to the discussion in response to this point.

15. Figure 4 suggests that gene regulation most relevant to disease may not be driven primarily by bulk expression levels, but instead by other regulatory mechanisms such as splicing. Indeed, eGenes linked to EyeG2P disease genes show lower coefficients of variation and smaller absolute effect sizes. While the authors interpret these findings as consistent with the general observation that common eVariants tend to have smaller effects than rare ones, their additional claim of selective bias against rare eVariants in eye disease genes is overstated without direct population-genetic evidence (e.g., constraint metrics, allele frequency depletion analyses, evidence that depletion is specific to eyeG2P genes vs matched random or permuted control set)

We agree with the reviewer that additional analyses would be required to substantiate this observation and have reworded the discussion to ensure that this point is clear.

16. The rare variant analysis includes splicing and allelic imbalance outliers, yet the study only assessed expression QTLs. Why were sQTLs and ASE not evaluated? If the goal is to assess the power and contribution of rare variant analyses to transcriptomic regulation, then eQTLs, sQTLs, and ASE should all be systematically compared. It would be valuable to quantify how common versus rare variants mediate transcriptomic variation (expression, splicing, allelic imbalance) and to report the proportion of variance explained in each category, as this would provide a clearer picture of their relative impact on disease mechanisms.

Large scale splicing QTL analyses for the retina are yet to be described for the retina or RPE. There is high complexity associated with sQTL analyses, including but not limited to accurate clarification of the retinal transcriptome (at isoform level) and additional functional validation of candidate variants. Extended functional follow-up has increased importance with the lack of comparisons that can be performed against independent datasets and as a result is out of scope of the current manuscript, although we see this as an important area for future collaboration and will be enabled by access to our dataset through the European-Genome phenome Archive.

(2) Additional clarifications / justifications of approaches taken

1. The authors have mentioned they found 18,891 and 13,214 genes expressed at moderate (TPM >1) and high (TPM > 5) levels in both the tissues. What genes and TPM filter were used for differential expression analysis? How was the data normalized and what covariates were used for correcting for hidden factors and

batch effects as the NSR samples RNA sequencing was done in 5 batches and RPE samples in 3 batches?

Please see our response to major comment #4 which sets out the context for these analyses. Our analysis of differentially expressed genes included all genes in NSR and RPE where mean expression > 1 TPM and at least 20% samples had a read count of ≥ 10 ($n = 17,751$ genes). We utilised the *deseq* R package and corrected for age and sex as covariates. We were unable to correct for batch, as there was a colinear relationship between batch and tissue. To ensure repeatability of the trends observed, we replicated differential expression analysis using a different bioinformatics method (*edgeR*), which implements a trimmed mean of M-values (TMM) normalisation approach and identified 14,913 differentially expressed genes (FDR > 0.05). The findings from *edgeR* demonstrated a similar split between upregulated and downregulated genes in the RPE compared to the NSR when compared to *deseq*.

2. The authors state that expression variability was significantly higher in RPE compared to NSR across genes expressed at low, moderate, and high levels ($p < 2.2 \times 10^{-16}$; Fig. 1B). Could the authors clarify which statistical test was performed to assess this difference? Specifically, was this a group comparison across samples, or a paired analysis between matched RPE and NSR samples from the same individuals?

An independent t-test was used to compare the mean coefficient of variation for NSR samples to RPE samples. The coefficient of variation is a measure of expression variability across samples, calculated by dividing the standard deviation over the mean expression (per gene). Therefore, this was a group comparison across samples.

We have refined this analysis exclusively for samples with data for both tissues ($n = 158$) and repeated the expression variability comparison between NSR and RPE. An independent t-test confirmed that there is still significantly higher expression variability in RPE compared to NSR ($t = 36.39$, $p\text{-value} < 2.2E-16$).

3. Median age 71 [64,77], Median PMI 40 [32,44], Male predominance [63.7%] presents issues when comparing to Gtex or EyeGex. Male-female differences in gene expression are well documented for various tissues.

We agree with this point and have added to the discussion to enable increased context for interpretation of the results presented.

4. The authors have not mentioned anything about the ancestry of the samples. The samples are presumably of European ancestry. Was ancestry check done on the samples before running the QTL analysis?

We thank the reviewer for raising the importance of taking into account the genetic ancestry of the cohort. We inferred the genetic ancestry of the METR-GT samples using *somalier* (<https://github.com/brentp/somalier/wiki/ancestry>), using samples from the 1000 Genomes Project as reference for each super-population ancestry group. The predicted genetic ancestry of all donors in our cohort was European, and we have added this to the methods, results (**Supplementary figure 3**) and discussion.

5. For the WGS sample level quality control the authors found outliers. How many samples were finally used for the QTL analysis for NSR and RPE. The authors also need to provide QC plots to show how they identified the outlier's samples.

We thank the reviewer for highlighting this. The number of samples reported in the manuscript was post-QC, and we have now clarified this in the text. We have also added **Supplementary table 3** and **Supplementary figure 2**, including descriptions of median coverage, number of reads with $Q > 30$, percentage of the genome with $> 15x$ coverage, uniformity of coverage, total number variants, total number of SNVs, transition/transversion ratio and heterozygous/homozygous ratio.

6. How did the author decide to use 30 PEERS factors in the QTL analysis. How many total PEERS were identified. Did the authors perform bench marking for the PEER factors to identify the QTLs.

For all the eQTL analysis we followed the methodology used by the GTEx consortium. Specifically, they recommend using 30 PEER factors for cohort sizes between 150 and 250 samples (<https://www.gtexportal.org/home/methods>). This has been added to the methods section of the manuscript.

7. The gene ontology results for the differentially expressed genes in Supplementary table 3 and Table 4 has redundant GO terms and genes. The redundant terms can be summarized based on semantic similarity or hierarchical relationships.

We agree that a more sophisticated approach to remove redundant terms adds value and interpretability to this analysis. In response to this comment, we have updated this analysis and utilised *rrvgo* to cluster GO terms based on semantic similarity and choose a single representative term to describe the whole group. This has been added to the methods, results (**Figure 1C**) and discussion.

10. Please clarify the statistical basis for classifying a gene as having 'no METR eQTL'. Is this defined solely by the absence of overlapping eQTLs under an FDR threshold of 0.05?

The statistical basis for classifying a gene as having 'no METR eQTL' is that the AMD gene in question was not associated with any eVariant in our cohort under an FDR threshold of 0.05. We have clarified this in the table legend.

12. What was the r^2 scores in LD for variants identified in the eQTLs reported in Table 1.

We used an r^2 threshold of 0.8 and have included this in the table legend.

13. Lines 110–111: The text states that 13 previously reported AMD-associated eQTLs were replicated (Table 1). However, the legend of Table 1 specifies that 15/26 previously reported eQTLs were replicated or in high LD. This discrepancy

should be clarified — is the correct number 13 or 15? Please ensure consistency between the main text and table legend.

We have added additional clarity to this statement in the table legend to increase interpretability. Specifically, we state that 15 eQTLs were either replicated ($n = 13$) or in high LD ($r^2 > 0.8$; $n=2$) with METR-eQTLs.

14. The authors have compared METR-eVariants to cell-type agnostic cis-candidate regulatory elements (cCRE) available through ENCODE and found METR-eVariants were enriched in promoters and proximal enhancers compared to control variants matched for allele frequency and gene density. The variants which are reported in Table 1 for AMD GWAS they were enriched in which regions.

We have added a summary of the overlap analyses for eQTLs described in Table 1 for ENCODE cCREs, and datasets published by Cherry et al, and Wang et al. This has been added text to the methods, results and discussion. Please also see response to reviewer #2 comment #3.

(3) eOutlier event analysis

17. Please define the method or criteria used to classify eOutliers as ‘co-occurring’ with SVs/CNVs. For example, is this defined as an outlier expression event for a gene in an individual overlapping with a rare variant in the same gene in the same individual? Please clarify the method, definition, and thresholds.

An eOutlier was defined as co-occurring with an SV/CNV if it overlapped with the coding sequence of the gene identified as an expression outlier in the same individual. We used the SV and CNV caller from Illumina DRAGEN v4.0.3 to identify structural and copy-number variants respectively, GENCODE v38 for the CDS annotations and *bedtools intersect* to identify overlapping SVs/CNVs. We have added clarity about this to the methods and ensured that this is captured in **Supplementary figure 12**.

18. Lines 175–176: Please define what constitutes an ‘outlier event.’ Is this defined as a gene–individual pair with expression significantly deviating from the expected distribution? Additionally, clarify the cutoff used — was this based on Z-score thresholds, residual modeling, or strictly on an adjusted p-value < 0.05 from the aberrant expression module?

An outlier event is defined as a gene–individual pair with expression significantly deviating from the expected distribution. We used DROP to identify outlier events, which uses the previously published OUTRIDER tool (<https://pmc.ncbi.nlm.nih.gov/articles/PMC6288422/>). We have included all parameters that were used in the methods. In brief, the model to detect outliers is optimised for each gene and the authors recommend that a fixed Z-score threshold is not required.

19. Consider defining an explicit Z-score threshold for calling outlier events, rather than relying only on adjusted p-values. For example, using both $p_{adj} < 0.05$ and $|Z| >$

2 could help benchmark sensitivity (e.g., at ~1:5 outlier-to-gene ratio) and make the definition of outlier events more interpretable. Additionally, it would be valuable to present enrichment analyses by minor allele frequency bins, stratified across variant classes (SNVs, indels, CNVs, SVs), to better contextualize how rare versus common variants contribute to outlier expression.

We understand that an explicit Z-score may help make the definition of outlier events more interpretable, however the strength of the OUTRIDER model is the ability to quantify the significance of an outlier event with a p-value, which can then be adjusted to control the false discovery rate. Please also see response to minor comment #18.

20. Table 2: The phrasing ‘median number of impacted genes per sample across expression, splicing, and allelic balance’ is not very intuitive in meaning. Did the authors first check whether specific samples were outliers that might be disproportionately driving these events?

We have reworded this phrasing.

21. Line 194: The phrasing ‘METR outliers’ should be revised to ‘outlier events.’ Referring to them simply as ‘outliers’ risks implying that an entire sample is an outlier, whereas the analysis is performed at the level of specific gene–sample outlier events.

We have reworded this to ‘eOutlier events’ throughout the manuscript.

22. Lines 197–199: Please specify the method used to link unexplained eOutlier events with cCREs. For example, was the approach: for each unexplained outlier event, determine whether the same individual carries a rare variant (SNV, indel, SV, CNV) within 10 kb of the gene body that overlaps an eye-specific cCRE (Cherry et al., 2020)?

For each unexplained outlier event we determined whether the same individual carried a rare SNV/indel or structural/copy-number variant within 10 kb of the gene body that overlapped an eye-specific cCRE from Cherry et al. (2020). This approach is described in the methods section of the paper and in **Supplementary figure 12**.

23. Lines 782–797: The definition of shared eQTLs relies on exact matches of variant ID (CHR_POS_REF_ALT, GRCh38) and Ensembl gene ID (without version). Please clarify how harmonization was performed across datasets to ensure consistency in variant calling and gene annotation. For EyeGEx replication, you extracted top eQTLs and checked for replication or high LD ($r^2 > 0.8$) using LDlinkR — it would help to specify whether allele harmonization (strand/orientation) was also conducted.

We agree that harmonisation across datasets when comparing eQTLs/eGenes is essential. EyeGex, Strunz et al. (2020) and Orozco et al (2020) provided gene IDs from ENCODE and dbSNP variant IDs. Therefore, we converted all our variant IDs from the standard format (chr_pos_ref_alt) to dbSNP IDs (using build release 156) prior to the comparison with other datasets. To compare eGenes, we used Ensembl

gene IDs without the version suffix. We have added these approaches to our methods section.

24. The manuscript should explain how non-genetic drivers of widespread extreme expression were handled — for example, by removing individuals with outliers in >50 genes from downstream analysis! The authors note that excluding such individuals improved rare variant enrichments, but it's unclear how this threshold was chosen and whether the results are robust to alternative cutoffs. Additionally, do the authors have replication for the identified eOutliers? The methods section states that enrichment was assessed only in genes with at least one outlier and one control, and that relative risk was estimated for the presence of a nearby rare variant given outlier status (with 95% Wald CIs). Explicitly describing the replication strategy, alongside this enrichment framework, would make the conclusions more convincing.

We agree with the reviewer that providing some understanding to the reader about the expectations of outlier events is important, and have added reference and details from a recent study using GTEx data to show that our values of eOutliers are in line with expectations. We do see natural replication of some eOutlier variants when more than one individual in the cohort shares the same rare variant, for example the *CAND2* variant discussed in **Figure 7**. In addition, we have added functional analyses using a dual reporter luciferase assay to demonstrate and confirm the impact of variants impacting the suspected *CAND2* promoter region (**Figure 7**) and provide proof-of-principle that the approaches undertaken identify genuine non-coding candidate variants that disrupt gene expression.

25. Please define explicitly what constitutes an 'outlier' in this ASE analysis. Does this include both underexpression and overexpression events, or only one direction? The text suggests expectations for both (minor allele underexpressed in underexpression outliers, overexpressed in overexpression outliers), but the definition of outlier status and how it was assigned (e.g., based on Z-scores, adjusted p-values, thresholds) should be clarified.

We considered allele-specific expression outliers to be gene-sample pairs where the adjusted p-value from the DROP mono-allelic expression module was lower than 0.05. This included both underexpression and overexpression events, as shown in the 'fold change' column in original **supplementary table 5** (now **supplementary table 6**).

26. The DROP workflow is currently used for rare disease diagnosis purposes so how is this method useful in complex diseases such as AMD? It is not clear what are the author's trying to convey with the list provided in Supplementary table 5.

Whilst the DROP workflow has utility in rare disease diagnostics, it is not restricted to this context. The gene outliers along with variants prioritised by our approach provided in original **Supp table 5** (now **Supp table 6**) are a series of candidates that may have diverse utility to the community for further functional follow-up. For example, they could be utilised to help identify and inform investigations of functional hotspots and/or clarify the regulatory roles of non-coding regions in NSR and RPE.

27. Authors have not explained what expression outliers are in Result section.

A definition has been added to the results section.

28. The methods Figure 1, Supplementary figure 3, Figure 6 and 7: the font size is too small and hard to read. The figures also get blurred when tried to enlarge.

We have made some changes to the identified figures to assist readability and will work further with the editorial team should any additional modifications need to be made.